# Improvement of muscle strength in a mouse model for congenital myopathy treated with HDAC and DNA methyltransferase inhibitors

**Alexis Ruiz[1], Sofia Benucci[1], Urs Duthaler[2], Christoph Bachmann[1], Martina Franchini[1], Faiza Noreen[3], Laura Pietrangelo[4], Feliciano Protasi[4], Susan Treves[1,5], Francesco Zorzato[1,5]***

[1]Neuromuscular Research Group, Departments of Neurology and Biomedicine, Basel University Hospital, Basel, Switzerland; [2]Division of Clinical Pharmacology & Toxicology, Department of Biomedicine, University and University Hospital Basel, Basel, Switzerland; [3]Genome plasticity group, Department of Biomedicine, University of Basel, Basel, Switzerland; [4]CAST, Center for Advanced Studies and Technology & DMSI, Dept. of Neuroscience, Imaging and Clinical Sciences, Univ. G. d'Annunzio, Chieti, Italy; [5]Department of Life Science and Biotechnology, University of Ferrara, Ferrara, Italy

*For correspondence: fzorzato@usb.ch

Competing interest: The authors declare that no competing interests exist.

**Abstract** To date there are no therapies for patients with congenital myopathies, muscle disorders causing poor quality of life of affected individuals. In approximately 30% of the cases, patients with congenital myopathies carry either dominant or recessive mutations in the ryanodine receptor 1 (*RYR1*) gene; recessive *RYR1* mutations are accompanied by reduction of RyR1 expression and content in skeletal muscles and are associated with fiber hypotrophy and muscle weakness. Importantly, muscles of patients with recessive *RYR1* mutations exhibit increased content of class II histone deacetylases and of DNA genomic methylation. We recently created a mouse model knocked-in for the p.Q1970fsX16+ p.A4329D RyR1 mutations, which are isogenic to those carried by a severely affected child suffering from a recessive form of RyR1-related multi-mini core disease. The phenotype of the RyR1 mutant mice recapitulates many aspects of the clinical picture of patients carrying recessive *RYR1* mutations. We treated the compound heterozygous mice with a combination of two drugs targeting DNA methylases and class II histone deacetylases. Here, we show that treatment of the mutant mice with drugs targeting epigenetic enzymes improves muscle strength, RyR1 protein content, and muscle ultrastructure. This study provides proof of concept for the pharmacological treatment of patients with congenital myopathies linked to recessive *RYR1* mutations.

## Editor's evaluation

The paper describes improvement in muscle phenotype of a congenital myopathy mouse model by a combined treatment with pharmacological inhibitors of Class IIa histone deacetylases and DNA methylases. The paper demonstrates in principle that there are treatment avenues to pursue but their application could be limited as phenotypic rescue appears to be restricted to particular muscle fiber types.

## Introduction

Skeletal muscle contraction is initiated by a massive release of $Ca^{2+}$ from the sarcoplasmic reticulum (SR) via the opening of the ryanodine receptor 1 (RyR1), a calcium release channel, which is localized in the SR terminal cisternae (*Rios and Pizarro, 1991*; *Endo, 1977*; *Fleischer and Inui, 1989*). The signal causing the opening of the RyR1 is the depolarization of the sarcolemmal membrane, which is sensed by voltage-dependent L-type $Ca^{2+}$ channels (dihydropyridine receptor [DHPR]) located in invaginations of the sarcolemma referred to as transverse tubules (TTs) (*Rios and Pizarro, 1991*; *Franzini-Armstrong and Jorgensen, 1994*). Communication between DHPR and RyR1 occurs in specialized intracellular junctions between TT and SR called $Ca^{2+}$ release units (CRUs). Skeletal muscle relaxation is brought about by SR $Ca^{2+}$ uptake via the activity of the sarco(endo)plasmic reticulum CaATPAses (SERCA) (*MacLennan, 2000*). Dis-regulation of $Ca^{2+}$ signals due to defects in key proteins (RyR1 and DHPR) involved in excitation-contraction (EC) coupling (ECC) is the under-lying feature of several neuromuscular disorders (*Jungbluth et al., 2018*; *Treves et al., 2008*; *Lawal et al., 2020*). In particular, mutations in *RYR1*, the gene-encoding RyR1, are causative of malignant hyperthermia (MH; MIM #145600), central core disease (CCD; MIM #11700), specific forms of multi-minicore disease (MmD; MIM # 255320), and centronuclear myopathy (*Jungbluth et al., 2018*; *Treves et al., 2008*; *Lawal et al., 2020*; *MacLennan and Philips, 1992*). A great deal of data has shown that *RYR1* mutations result mainly in four types of channel defects (*Treves et al., 2008*). One class of mutations (dominant, MH-associated) causes the channels to become hypersensitive to activation by electrical and pharmacological stimuli (*MacLennan and Philips, 1992*). The second class of *RYR1* mutations (dominant, CCD-associated) results in leaky channels leading to depletion of $Ca^{2+}$ from SR stores (*Treves et al., 2008*; *Lawal et al., 2020*). A third class of *RYR1* mutations also linked to CCD causes EC uncoupling, whereby activation of the voltage sensor $Ca_v1.1$ is unable to cause release of $Ca^{2+}$ from the SR (*Avila et al., 2003*). The fourth class comprises recessive mutations, which are accompanied by a decreased content of mutant RyR1 channels on SR membranes (*Wilmshurst et al., 2010*; *Monnier et al., 2008*; *Zhou et al., 2007*; *Zhou et al., 2013*).

Patients with congenital myopathies such as MmD carrying recessive *RYR1* mutations belonging to class 4 channel defects, typically exhibit non-progressive proximal muscle weakness (*Jungbluth et al., 2005*; *Klein et al., 2012*).

This reduced muscle strength is consistent with the lower RyR1 content observed in adult muscle fibers that should result in a decrease of $Ca^{2+}$ release from the SR (*Wilmshurst et al., 2010*; *Monnier et al., 2008*; *Zhou et al., 2007*; *Zhou et al., 2013*; *Jungbluth et al., 2005*). The decrease of RyR1 expression is also associated with moderate fiber atrophy, which may additionally contribute to the decrease of muscle strength. In addition to the depletion of RyR1 protein, muscles of patients with recessive *RYR1* mutations exhibit striking epigenetic changes, including altered expression of microRNAs, an increased content of HDAC-4 and HDAC-5, and hypermethylation of more than 3600 CpG genomic sites (*Zhou et al., 2006*; *Rokach et al., 2015*; *Bachmann et al., 2019*). Importantly, in muscle biopsies from four patients, hypermethylation of one of the internal *RYR1* CpG islands correlated with the increased levels of HDAC-4 and HDAC-5 (*Rokach et al., 2015*).

In order to study in more detail the mechanism of disease of recessive *RYR1* mutations, we developed a mouse model knocked in for two mutations, isogenic to those identified in a severely affected child with recessively inherited MmD (*Klein et al., 2012*). Such a mouse model carries the p.Q1970fsX16 + p.A4329D RyR1 mutations (*Elbaz et al., 2019*) and will henceforth be referred to as double heterozygous (dHT). Characterization of the muscle phenotype of dHT mice demonstrated that it faithfully recapitulates not only the physiological and biochemical changes, but also the major muscle epigenetic signatures observed in muscle biopsies from MmD patients. In the present study we treated dHT mice with drugs targeting epigenetic enzymes and evaluated the physiological effects of the treatment on muscle function as well as on muscle structure. Our results show that treatment of dHT mice with drugs targeting epigenetic enzymes rescues muscle strength, increases RyR1 protein content, and improves muscle morphology, that is, the treatment partially rescues CRUs (the intra-cellular sites containing RyR1) and mitochondria. This study provides proof of concept for the treatment of patients with congenital myopathies linked to recessive *RYR1* mutations, with small molecules inhibiting DNMT and histone deacetylases.

## Results

## Effect of TMP269 and 5-aza-2-deoxycytidine on the in vivo muscle phenotype of dHT mice

We first examined the pharmacokinetics and bio-distribution of TMP269, the class IIa HDAC inhibitor we selected for this study. After intraperitoneal (i.p.) injection of 25 mg/kg body weight of TMP269 dissolved in polyethylenglycol 300 (PEG300) (500 µl/kg) and *N*-methyl-2-pyrrolidone (NMP) (250 µl/kg), blood and/or skeletal muscles were collected at different time points and the content of TMP269 was quantified by liquid chromatography tandem mass spectrometry (LC-MS/MS) (*Duthaler et al., 2019*). The peak blood concentration of TMP269 was achieved approximately 1 hr after injection. The circulating levels of TMP269 decay within 12 hr (*Appendix 1—figure 1A*). Importantly, the class II HDAC inhibitor diffuses into skeletal muscle (*Appendix 1—figure 1B*), and as expected, its concentration profile in skeletal muscle follows that observed in blood. Although the level of TMP269 accumulating in skeletal muscle is lower compared to that present in blood, its concentration in muscle is adequate to induce an inhibitory effect on class IIa HDACs activity (*Choi et al., 2018*). An identical protocol was used to monitor the optimal dose of 5-aza-2-deoxycytidine (5-Aza), an FDA-approved DNA methyltransferase (DNMT) inhibitor (*Kaminskas et al., 2005*). Administration of TMP269 + 5-Aza for 15 weeks resulted in hypomethylation of 165 protein-coding genes (*Supplementary file 1*) in soleus muscles. Gene Ontology analysis showed that most of the hypomethylated genes belong to pathways involved in gene transcription, kinase activity, and membrane targeting (*Appendix 2—figure 1*). Importantly, administration of TMP269 + 5-Aza for 15 weeks increases the acetylation of Lys residues (*Appendix 2—figure 2A* and *Appendix 2—figure 2—source data 1*; *Appendix 2—figure 2—source data 2*; *Appendix 2—figure 2—source data 3*; *Appendix 2—figure 2—source data 4*; *Appendix 2—figure 2—source data 5*) and of H3K9 (*Appendix 2—figure 2B and D* and *Appendix 2—figure 2—source data 1*; *Appendix 2—figure 2—source data 2*; *Appendix 2—figure 2—source data 3*; *Appendix 2—figure 2—source data 4*; *Appendix 2—figure 2—source data 5*) in total homogenates from *flexor digitorum brevis* (FDB) fibers isolated from wild type (WT) and dHT mice, compared to that observed in fibers from vehicle-treated WT and dHT mice. This result unequivocally indicates that the inhibition of the deacetylation activity of class IIa HDACs

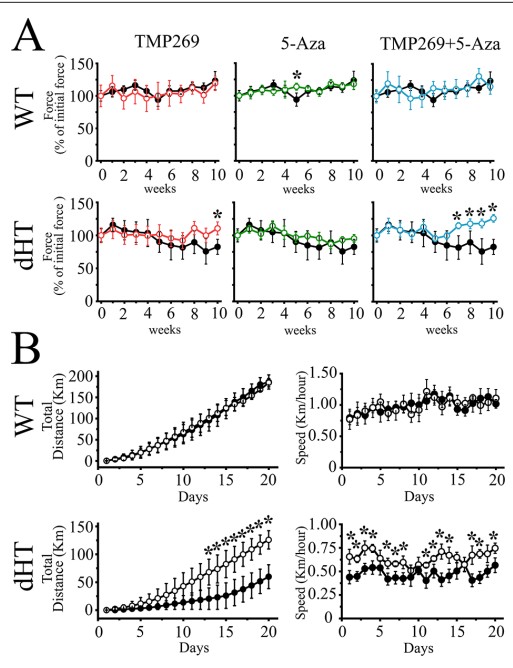

**Figure 1.** Treatment of double heterozygous (dHT) mice with TMP269 + 5-aza-2-deoxycytidine (5-Aza) improves in vivo muscle function as assessed using the grip strength test and voluntary running wheel. (**A**) Forelimb (two paws) grip force measurement of wild type (WT) (upper panels) and dHT (lower panels) mice treated with vehicle (WT, n = 9; dHT, n = 10), TMP269 (WT, n = 11; dHT, n = 6), 5-Aza (WT, n = 5, dHT, n = 10), and TMP269 + 5-Aza (WT, n = 10, dHT, n = 13). Grip strength was performed once per week during a period of 10 weeks. Each symbol represents the average (± SD) grip force obtained in the indicated number (n) of mice. Grip force (Force) values obtained on the first week were considered 100%. Black symbols, vehicle-treated mice, colored symbols, drug-treated mice. Statistical analysis was conducted using the Mann-Whitney test. *p < 0.05. (**B**) Spontaneous locomotor (dark phase) activity (left panel) and total running speed (right panel) measured over 20 days in 21-week-old dHT and WT littermates mice treated with vehicle or TMP269 + 5-Aza. Data points are expressed as mean (± SD; n = 4–5 individual mice). *p < 0.05 (Mann-Whitney test). The exact p value for day 20 is given in the text.

The online version of this article includes the following source data and figure supplement(s) for figure 1:

**Source data 1.** Grip strength in double heterozygous (dHT) mice is improved after 10 weeks of treatment with TMP269 + 5-aza-2-deoxycytidine (5-Aza).

**Figure supplement 1.** Grip strength in double heterozygous (dHT) mice is improved after 10 weeks of treatment with TMP269 + 5-aza-2-deoxicytidine (5-Aza).

by TMP269 occurs in the nuclei. In addition, *Appendix 2—figure 2* also shows that in the absence of TMP269 + 5-Aza treatment, the level of H3K9 acetylation in myonuclei from FDB fibers isolated from dHT mice is 50% lower compared to WT. This result is consistent with: (i) higher deacetylation activity by class IIa HDACs in vehicle-treated dHT mice compared to TMP269 + Aza-treated dHT mice and WT mice; (ii) enrichment of class II HDACs in the nuclei of dHT mice. Furthermore, these results confirm that the drugs reach skeletal muscles where they exerted their biological function.

We next conducted a series of experiments to determine how the drug treatment affects in vivo skeletal muscle function. To this end we injected 6-week--old WT and dHT mice with vehicle alone (PEG300 + NMP), with TMP269 alone (25 mg/kg), 5-Aza alone (0.05 mg/kg), or with the two drugs combined on a daily basis and investigated their effects on the in vivo skeletal muscle phenotype by analyzing forelimb grip force using a grip strength meter. Administration of each drug singly, namely TMP269 or 5-Aza alone, does not induce any change in the grip strength of WT (*Figure 1A*, top left and middle panels). The combined drug treatment did not affect the grip strength of WT mice (*Figure 1A*, top right panel). In dHT mice, TMP269 alone causes a small but significant increase of grip strength after 10 weeks of treatment (*Figure 1A*, low left panel). However, the increased grip strength in dHT mice (*Figure 1A*, lower right panel) was more evident by the combined drug treatment. In particular, this effect became apparent 4/5 weeks after starting the drug treatment and peaked at 10 weeks. The combined drug treatment rescues approximately 20% of muscle grip strength in dHT mice (see *Figure 1—figure supplement 1* for raw data on the grip strength of individual dHT mice before and after treatment with TMP269 + 5-Aza). Based on these results, we performed all subsequent experiments only with the combined drug treatment (i.e. TMP269 + 5-Aza).

We next assessed in vivo muscle function of WT and dHT vehicle-treated or drug-treated mice using the voluntary running wheel. We calculated the total running distance of WT (*Figure 1B*, top panels) and dHT (*Figure 1B*, bottom panels) mice injected with vehicle and compared it to that of mice treated for 15–18 weeks with TMP269 + 5-Aza. Three weeks of training improved running performance in both mouse groups. Nevertheless, on day 20 the total running distance of WT mice injected with vehicle alone was approximately 70% greater compared to that of vehicle-treated dHT mice: the total running distance of vehicle-treated WT and dHT mice was 186.27 ± 16.70 km n = 4 vs. 59.93 ± 21.38 km n = 5, respectively (mean ± SD, Mann-Whitney two-tailed test, calculated over the 20 days of running, *p < 0.05). On the other hand, treatment of dHT mice with TMP269 + 5-Aza has a remarkable effect on the total running distance (*Figure 1B*, lower panel). The beneficial effect begins 1 week after treatment commencement, and on day 20 the total running distance achieved by dHT mice injected with TMP269 + 5-Aza was two times higher compared to that covered by dHT mice injected with vehicle alone (*Figure 1B*, lower panel; Mann-Whitney two-tailed test; on day 20, p = 0.041): the total running distance for vehicle-treated and drug-treated dHT mice was 59.93 ± 21.38 km n = 5 and 125.90 ± 16.39 km n = 5, respectively (mean ± SD, Mann-Whitney two-tailed test, calculated over the 20 days of running *p < 0.05). The longer running distance was also associated with an increased median cruise speed of the drug-treated dHT mice compared to vehicle-treated dHT mice (*Figure 1B*, right panels) (Mann-Whitney two-tailed test, calculated over the 20 days running period, *p < 0.05). The epigenetic modifying drugs most likely affect a number of genes, which in turn leads to an improvement of the in vivo muscle performance of the dHT mice; the latter effect may result from an improvement of the mechanical properties of skeletal muscles and/or by an influence of the drugs on the metabolic pathways of muscles. In the next set of experiments, we investigated the mechanical properties of intact *extensor digitorum longus* (EDL) and soleus muscles from WT and dHT mice after injection of vehicle and of TMP269 + 5-Aza.

## Effect of TMP269+5-Aza treatment on isometric force development in muscles from WT and dHT heterozygous mice

EDL and soleus muscles isolated from mice treated for 15 weeks with vehicle or TMP269 + 5-Aza were stimulated with a single 15 V pulse of 1.0 ms duration (*Figure 2A, C, E and G*) or by a train of pulses delivered at 150 Hz for 400 ms (EDL, *Figure 2B and D*) or 120 Hz for 1100 ms (soleus, *Figure 2F and H*) to obtain maximal tetanic contracture. The averaged specific twitch peak force induced by a single action potential in EDL from dHT mice injected with vehicle alone was approximately 37% of that obtained from EDLs from WT mice ($64.92 \pm 13.93$ mN/mm$^2$, n = 10 vs. $171.24 \pm 29.32$** mN/mm$^2$, n = 8, respectively; mean ± SD; ANOVA followed by the Bonferroni post hoc test **p < 0.01;

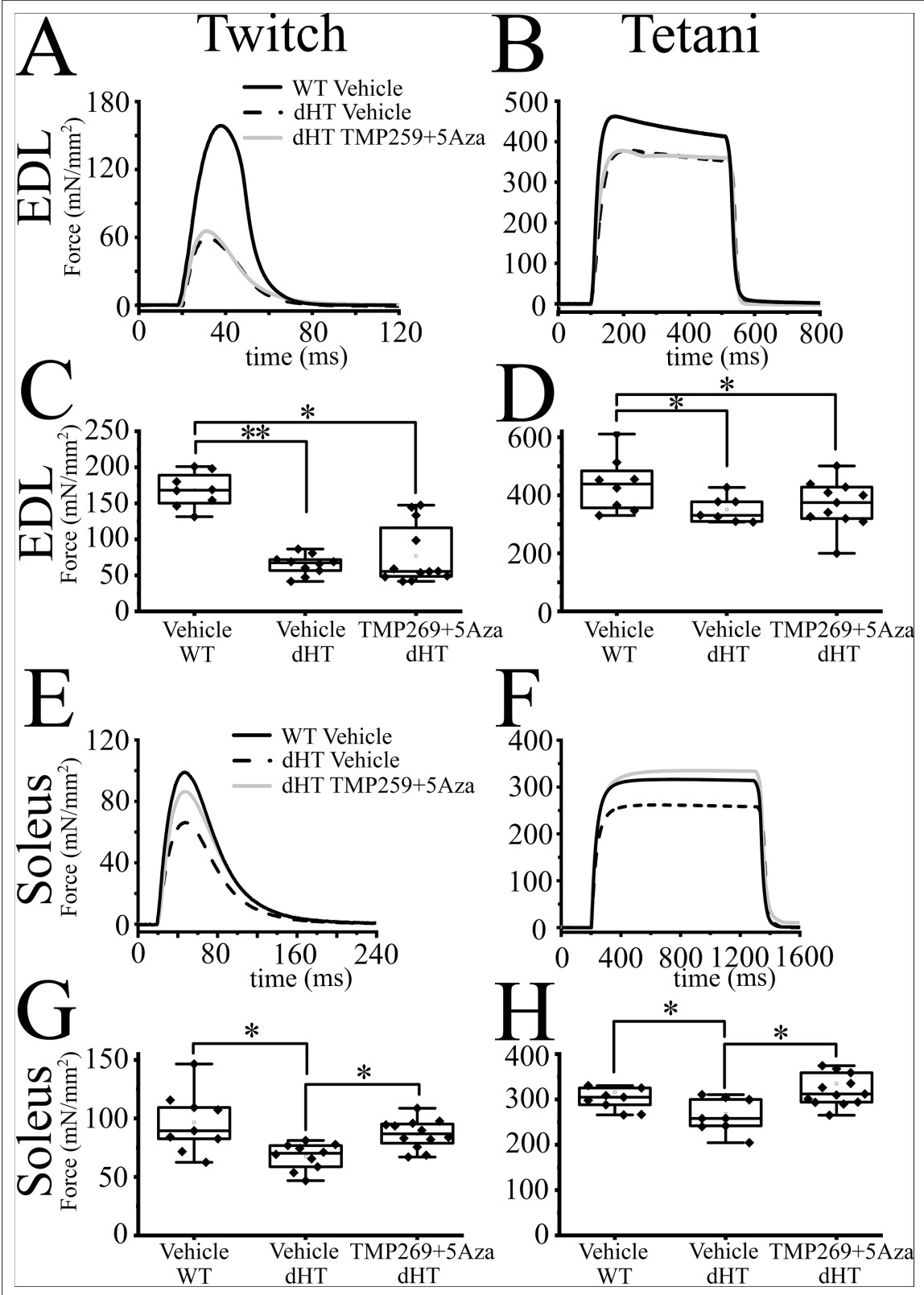

**Figure 2.** The mechanical properties of double heterozygous (dHT) treated with TMP269 + 5-aza-2-deoxycytidine (5-Aza) improve after 15 weeks of treatment. Mechanical properties of *extensor digitorum longus* (EDL) and soleus muscle from wild type (WT) and dHT mice treated with vehicle (WT, n = 8; dHT, n = 10) and dHT treated with TMP269 + 5-Aza, n = 13. (**A**) Representative traces of twitch and (**B**) maximal tetanic force in EDL (150 Hz) muscle from WT and dHT. Force is expressed as specific force, mN/mm². (**C**) Statistical analysis of force generated after twitch and (**D**) tetanic stimulation of

*Figure 2 continued on next page*

*Figure 2 continued*

isolated EDL muscle. Data points are expressed as Whisker plots (n = 8–13 mice). Each symbol represents the value from a muscle from a single mouse. (**E**) Representative traces of twitch and (**F**) maximal tetanic force (120 Hz) of soleus muscle from WT and dHT mice. (**G**) Whisker plots of force generated after twitch and (**H**) tetanic stimulation of isolated soleus muscles. Each symbol represents the value from a muscle from a single mouse (n = 8–13 mice). *p < 0.05; **p < 0.01 (ANOVA followed by the Bonferroni post hoc test). The exact p values are given in *Supplementary file 2*.

*Supplementary file 2*). The peak force developed after twitch stimulation of soleus muscles from dHT mice injected with vehicle alone was approximately 67% of that of obtained in soleus muscles from WT littermates injected with vehicle (67.55 ± 11.26 mN/mm², n = 10 vs. 96.58 ± 25.78* mN/mm², n = 8, respectively; mean ± SD; ANOVA followed by the Bonferroni post hoc test *p < 0.05; *Supplementary file 2*). While the combined drug treatment does not affect the force developed by EDL muscles, we found that it induces a 25% increase of the twitch force in soleus muscles from dHT mice compared to that obtained from vehicle-treated dHT mice (84.61 ± 14.06 mN/mm², n = 13 vs. 67.55 ± 11.26* mN/mm², n = 10, respectively; mean ± SD; ANOVA followed by the Bonferroni post hoc test *p < 0.05; *Figure 2G* and *Supplementary file 2*). We next examined the force developed during tetanic contractures of EDL and soleus muscles stimulated by a train of pulses delivered at 150 and 120 Hz, respectively. The maximal specific tetanic force developed in EDL muscles from dHT mice injected with vehicle was approximately 20% lower compared to that of EDL muscles from WT mice (373.76 ± 73.16* mN/mm², n = 10 vs. 452.97 ± 89.59 mN/mm², n = 8, respectively; mean ± SD; ANOVA followed by the Bonferroni post hoc test *p < 0.05; *Figure 2B and D* and *Supplementary file 2*). We found no effect of the combined drug treatment on the maximal specific force developed by EDL muscles isolated from dHT mice (*Supplementary file 2*). The maximal specific tetanic force generation observed in soleus muscles from dHT mice injected with vehicle was 13% lower compared to WT (276.29 ± 40.04* mN/mm², n = 10 vs. 315.86 ± 56.96 mN/mm², n = 8, mean ± SD; ANOVA followed by the Bonferroni post hoc test *p < 0.05) (*Figure 2F and H*). Contrary to what we observed in EDL muscles, the combined drug treatment fully rescues the maximal tetanic force of slow twitch muscles. Indeed, soleus from dHT mice treated with TMP269 + 5-Aza for 15 weeks displayed a maximal tetanic force which was 18% higher compared to that of soleus from dHT mice injected with vehicle (334.78 ± 65.74* mN/mm², n = 13 vs. 276.29 ± 40.04 mN/mm², n = 10, mean ± SD; ANOVA followed by the Bonferroni post hoc test *p < 0.05) (*Supplementary file 2*).

## Fiber type composition and minimal Feret's diameter of soleus muscles from dHT mice treated with TMP269+5-Aza

In this set of experiments, we examined whether the ergogenic effect associated with the inhibition of epigenetic modifying enzymes is linked to fast-to-slow fiber transition and/or to changes of minimal Feret's diameter. Treatment with TMP269 + 5-Aza causes no changes in the content of the fiber type composition of soleus muscles (*Figure 3A and B*, and *Supplementary file 3*). Similarly, the improved specific force cannot be attributed to a major shift of minimal Feret's fiber diameter distribution (*Figure 3C*).

## Effect of TMP269+5-Aza treatment on calcium transients in single FDB fibers from WT and dHT mice

We investigated resting [Ca²⁺] and calcium transients evoked either by a single pulse (*Figure 4A and B*) or by a train of action potentials (*Figure 4C and D*) in FDB fibers from WT and dHT mice treated for 15 weeks with vehicle or TMP269 + 5-Aza (4–6 mice per group). We used single FDB fibers (i) because they are a mixture of fast and slow twitch muscles and (ii) since intact single fibers from EDL and soleus muscles from 22-week--old mice are nearly impossible to obtain. We found that the resting [Ca²⁺] was similar in FDB fibers from WT and dHT vehicle- or drug-treated mice. The Fura-2 fluorescence values (F340/F380, mean ± SD) were 0.81 ± 0.09 (n = 75 fibers isolated from 4 mice), 0.77 ± 0.07 (n = 40 fibers isolated from 5 mice), and 0.81 ± 0.08 (n = 33 fibers isolated from 5 mice) in WT mice injected with vehicle, dHT mice injected with vehicle, and dHT mice treated with TMP269 + 5-Aza, respectively. We next investigated the Ca²⁺ transients in response to electrical stimulation in FDB fibers loaded with 10 µM of the low affinity Ca²⁺ indicator Mag-Fluo-4. In the presence of 1.8 mM Ca²⁺ in the extracellular solution, baseline fluorescence (mean ± SD fluorescence in arbitrary units) was similar in FDBs from vehicle-treated WT mice (1.13 ± 0.12, n = 91 fibers isolated from n = 4 mice), in

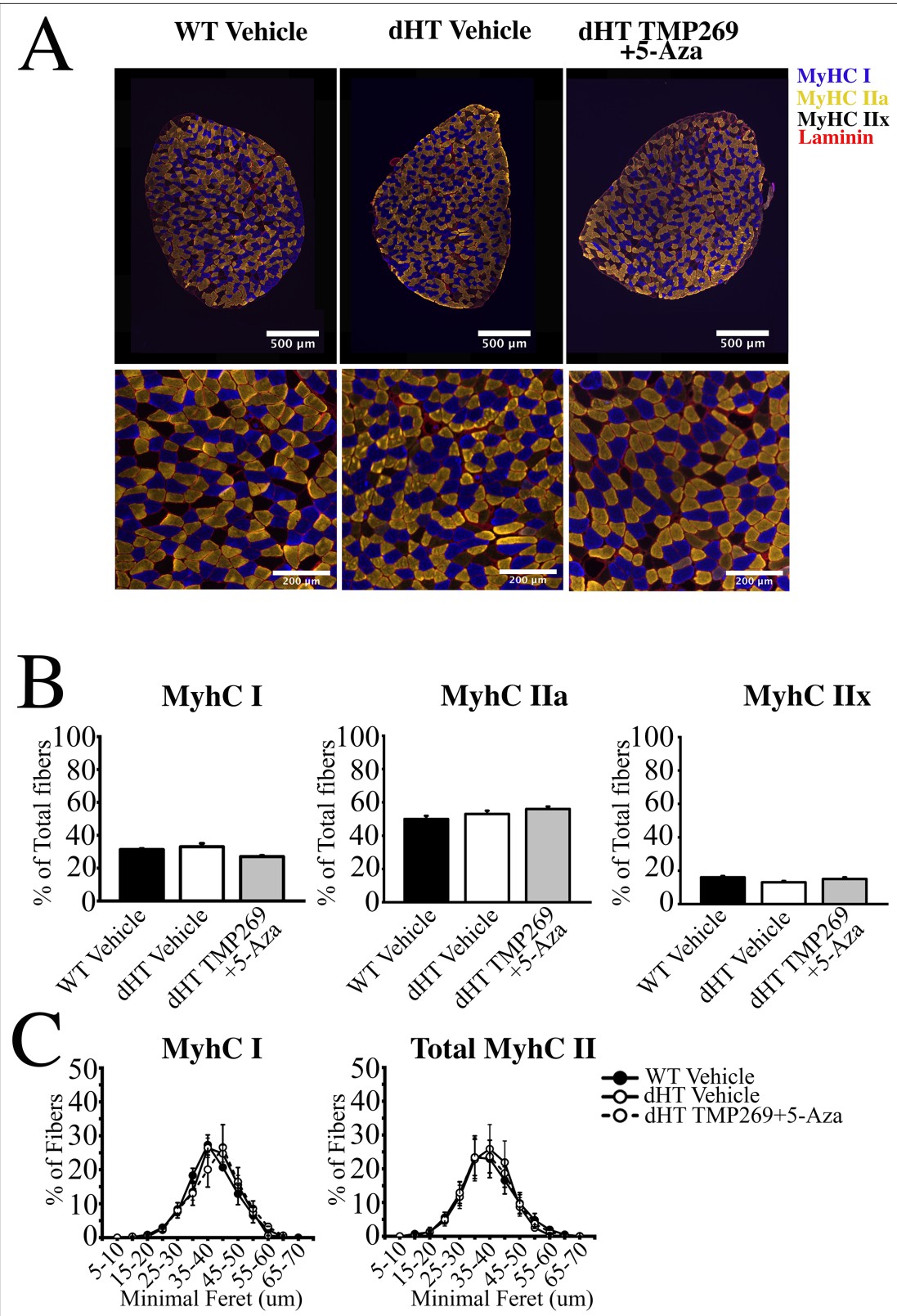

**Figure 3.** Histology of soleus muscles from TMP269 + 5-aza-2-deoxycytidine (5-Aza) and vehicle-treated double heterozygous (dHT) mice. (**A**) Analysis of soleus muscles from wild type (WT) (vehicle-treated) and dHT (vehicle and TMP269 + 5-Aza-treated) mice using monoclonal antibodies specific for myosin heavy chain (MyHC) isoforms. Frozen muscle sections were stained with anti-MyHC I antibodies (slow fibers, blue), anti-MyHC IIa antibodies (fast fibers, yellow), and counterstained with anti-laminin antibodies (red). MyHCIIx (fast fibers) are unstained. (**B**) Bar plots of fiber type composition

*Figure 3 continued on next page*

*Figure 3 continued*

of soleus muscles. Left, mean (%, ± SEM) MyHC I fibers, middle, mean (%, ± SEM) MyHC IIa fibers, right mean (%, ± SEM) MyHC IIx fibers. (**C**) Minimal Feret's distribution of type I and type II fibers. Data points are expressed as mean (± SEM). For WT vehicle treated a total of 2881 fibers from 3 mice were counted, for dHT vehicle treated a total of 2642 fibers from 3 mice were counted, for dHT TMP269 + 5-Aza a total of 2983 fibers from 3 mice were counted.

vehicle-treated dHT mice (1.14 ± 0.12, n = 110 fibers isolated from n = 6 mice), and drug-treated dHT mice (1.16 ± 0.10, n = 155 fibers isolated from n = 5 mice) (*Appendix 3—figure 1*). In the presence of 1.8 mM $Ca^{2+}$ in the extracellular solution, the average peak intracellular $Ca^{2+}$ transient induced by a single action potential in FDB fibers from dHT mice injected with vehicle is approximately 30% lower than that observed in fibers from WT mice (ΔF/Fo values were 0.98 ± 0.22, n = 110 fibers isolated from 6 mice vs. 1.38 ± 0.30, n = 91 fibers isolated from 4 mice, respectively; mean ± SD; *Figure 4A and B*, *Supplementary file 4*). Interestingly, treatment of dHT mice with TMP269 + 5-Aza causes a 23% increase of the peak $Ca^{2+}$ transient compared to that observed in dHT mice injected with vehicle (*1.21 ± 0.28, n = 155 fibers isolated from 5 mice vs. 0.98 ± 0.22, n = 110 fibers isolated from 6 mice, respectively; ΔF/Fo values are expressed as mean ± SD; ANOVA followed by the Bonferroni post hoc test *p < 0.05; *Figure 4A and B* and *Supplementary file 4*). In the presence of 1.8 mM $Ca^{2+}$ in the extracellular solution, the peak $Ca^{2+}$ transient evoked by a train of pulses delivered at 100 Hz in FDB fibers from dHT mice injected with vehicle is approximately 25% lower compared to that of FDB fibers from WT mice (*1.22 ± 0.22, n = 92 fibers isolated from 6 mice vs. 1.62 ± 0.21, n = 63 fibers isolated from 4 mice, respectively; ΔF/Fo values are expressed as mean ± SD; *Figure 4D* and *Supplementary file 4*). When dHT mice are treated with TMP269 + 5-Aza for 15 weeks, the summation of calcium transients induced by a train of supramaximal pulses is 16% higher compared to that of dHT mice injected with vehicle alone (*1.42 ± 0.23, n = 78 fibers isolated from 5 mice vs. 1.22 ± 0.22, n = 92 fibers isolated from 6 mice, respectively; ΔF/Fo values are expressed as mean ± SD; ANOVA followed by the Bonferroni post hoc test *p < 0.05; *Figure 4C and D* and *Supplementary file 4*). Altogether, the increase of the peak calcium transient after either a single action potential or a train of pulses is consistent, at least in part, with the ergogenic effects caused by the combined treatment with TMP269 + 5-Aza on dHT mice.

## TMP269+5-Aza rescues RyR1 expression in muscles from dHT mice

The results obtained so far indicate that treatment with TMP269 + 5-Aza exerts a beneficial effect preferentially on slow twitch muscles. Nevertheless, the genome-wide effects linked to the combined inhibition of class II HDACs and DNMT may affect a number of processes underlying muscle strength, making it difficult if not impossible to dissect the exact mechanisms underlying the improvement of slow twitch muscle function observed in dHT mice. However, based on our previous results, we postulate that the improvement of muscle strength observed in soleus muscles may be explained, at least in part, by an increase of the key proteins involved in skeletal muscle activation. *Figure 5A* shows *Ryr1* transcript expression in soleus muscles from WT and dHT mice. Treatment with vehicle alone does not rescue *Ryr1* expression (Mann-Whitney two-tailed test, WT vehicle vs. dHT vehicle, p = 0.039), however treatment with TMP269 + 5-Aza for 15 weeks causes a significant increase in *Ryr1* transcript levels (Mann-Whitney two-tailed test, dHT vehicle vs. dHT TMP269 + 5-Aza, p = 0.019). *Cacna1s* levels are not affected by vehicle or drug treatment. *Hdac4* transcript levels are increased in soleus muscles from dHT vehicle-treated mice, compared to vehicle-treated WT mice (*Figure 5A*, Mann-Whitney two-tailed test, WT vehicle vs. dHT vehicle, p = 0.041). *Hdac4* transcript levels decrease in muscles from TMP269 + 5-Aza-treated dHT mice compared to vehicle-treated dHT mice (*Figure 5A*, Mann-Whitney two-tailed test, dHT vehicle vs. dHT TMP269 + 5-Aza, p = 0.002). We also investigated RyR1 protein content in total homogenates from soleus muscles from WT and dHT mice. The RyR1 protein content of soleus muscles from dHT mice injected with vehicle is 46% lower compared to that of WT mice (*Figure 5B* and *Figure 5—source data 1–3*). The mean ± SD % intensity of the immunopositive band corresponding to RyR1 is 100% ± 7.30, n = 8 in WT vs. 54.93% ± 18.45, n = 6 in dHT, respectively (ANOVA followed by the Bonferroni post hoc test, *p = 0.039). The RyR1 protein content is partially re-established by treatment with TMP269 + 5-Aza. Indeed, the RyR1 protein content in drug-treated dHT mice increased and reaches a value of 88.03% ± 10.42*, n = 7, in contrast to the 54.93% ± 18.45, n = 6 found in vehicle-treated dHT mice (ANOVA followed by the Bonferroni post hoc test, *p = 0.037

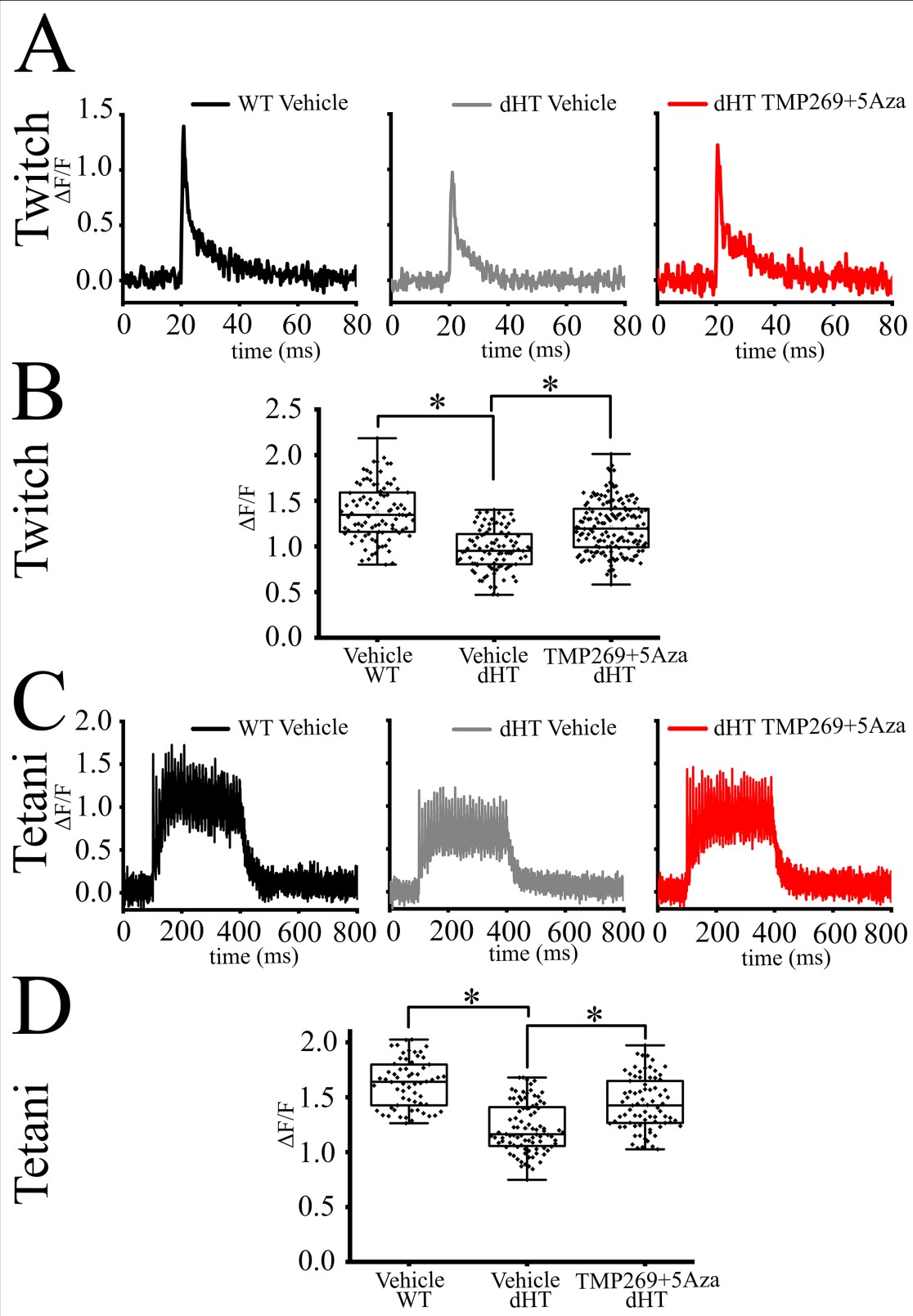

**Figure 4.** Electrically evoked peak Ca$^{2+}$ transients in muscle fibers from treated double heterozygous (dHT) compound heterozygous (dHT) mice was rescued by TMP269 + 5-aza-2-deoxycytidine (5-Aza) administration. Enzymatically dissociated *flexor digitorum brevis* (FDB) fibers dissected from 4 to 6 mice per group, were loaded with Mag-Fluo-4 and electrically stimulated by field stimulation. Black line, vehicle-treated wild type (WT), gray line, vehicle-treated dHT, red line, TMP269 + 5-Aza-treated dHT. (**A**) Representative Ca$^{2+}$ transient evoked by a single pulse (twitch) of 50 V with a duration

*Figure 4 continued on next page*

Figure 4 continued

of 1 ms. (**B**) Whisker plots of peak twitch. Each symbol represents results obtained from a single FDB fiber. (**C**) Representative Ca$^{2+}$ transient evoked by tetanic stimulation by a train of pulses delivered at 100 Hz for 300 ms. (**D**) Whisker plots of peak transient induced by tetanic stimulation. Each symbol represents results obtained from a single FDB fiber. *p < 0.05 (ANOVA followed by the Bonferroni post hoc test). The exact p values are given in **Supplementary file 4**.

for vehicle vs. TMP269 + 5-Aza-treated dHT mice; *Figure 5* and *Figure 5—source data 1–3*). This recovery of RyR1 is consistent with the in vivo and in vitro muscle phenotype amelioration induced by the combined drug treatment. We also measured the effect of the combined drug treatment on the content of other proteins involved in skeletal muscle ECC including SERCA1, SERCA2, calsequestrin-1, and JP-45. As shown in *Figure 5C* and *Figure 5—source data 4–11*, the muscle content of these proteins is unaffected by the drug treatment.

## Recovery of CRUs in soleus muscles from dHT after TMP269+5-Aza treatment

Skeletal muscle fibers from adult WT are usually characterized by a regular transverse pale-dark striations. Within the fiber interior CRUs, the SR-TT junctions containing RyR1 are uniformly distributed, and mostly placed at A-I band transition (when sarcomeres are relaxed), on both sides of the Z line. CRUs are formed by two SR terminal cisternae closely opposed to a central TT oriented transversally with respect to the longitudinal axis of the myofibrils. These CRUs are called triads. CRUs are often associated with a mitochondrion, to form functional couples (*Boncompagni et al., 2009*).

In soleus muscle fibers from vehicle-treated dHT mice (*Figure 6A and B*), CRUs present some abnormal features: the SR is often dilated (small arrows in *Figure 6A*) and at times they are incomplete, meaning they are formed by only two elements (dyads) (*Figure 6B*, black arrow). Quantitative electron microscopy (EM) analysis shows that in fibers of vehicle-treated dHT mice there are 40.3 ± 2.4 CRUs/100 µm$^2$, 14.2 ± 3.6% of them being dyads (*Table 1*, columns A–B). In soleus muscle fibers of dHT vehicle-treated mice, there are also regions with accumulated autophagic material (*Figure 6B*, empty arrow). In fibers from TMP269 + 5-Aza-treated dHT mice (*Figure 6C and D*), analysis of CRUs indicates a (at least partial) rescue of the ECC machinery. Dilated SR in triads (pointed by small arrows in *Figure 6A*) are practically absent (*Figure 6C*), and CRUs are more abundant and better preserved, meaning that they have the classic triad structure (three elements: two SR and one T-tubule, small arrows in *Figure 6D*). Quantitative analysis indicates that there are 46.9 ± 2.3 CRUs/100 µm$^2$, only 6.5% ± 0.6% of them being dyads (*Table 1*, columns A–B).

We also analyzed mitochondria and their association with CRUs (*Table 2*). In TMP269 + 5-Aza-treated dHT vs. vehicle-treated mice: (i) the number/area of mitochondria is significantly increased (71.0 ± 3.5 vs. 59.7 ± 2.9 mean ± SEM; *Table 2*, column A); (ii) the number of damaged mitochondria is slightly but significantly reduced (2.9 ± 0.6 vs. 3.2 ± 0.5 mean ± SEM; *Table 2*, column B), and finally (iii) mitochondria are more often correctly associated with CRUs to form functional couples (34.5 ± 2.3 vs. 26.9 ± 2.2 mean ± SEM; *Table 2*, column C).

## Discussion

Congenital myopathies are rare neuromuscular disorders, caused in approximately 30% of the patients, by mutations in the *RYR1* gene (*Jungbluth et al., 2018*; *Lawal et al., 2020*). Patients can present a variety of symptoms and phenotypes depending on whether the mutations are dominantly or recessively inherited. In particular, patients with recessive *RYR1* mutations often display involvement of extraocular muscles leading to ophthalmoplegia and or ptosis as well as involvement of respiratory muscles, often requiring assisted ventilation (*Jungbluth et al., 2018*; *Treves et al., 2008*; *Lawal et al., 2020*; *Wilmshurst et al., 2010*; *Jungbluth et al., 2005*). Children may also present physical abnormalities including club foot, scoliosis, facial dysmorphisms, winged scapula, and/or pectus excavatum (*Jungbluth et al., 2005*). Nevertheless, the most disturbing symptom present at infancy in children carrying recessive *RYR1* mutations are hypotonia and proximal muscle weakness. In the most severe cases, muscle weakness also impairs masticatory muscles causing dysphagia, whereby the affected infants are fed via percutaneous endoscopy gastrostomy, a procedure which influences the quality of

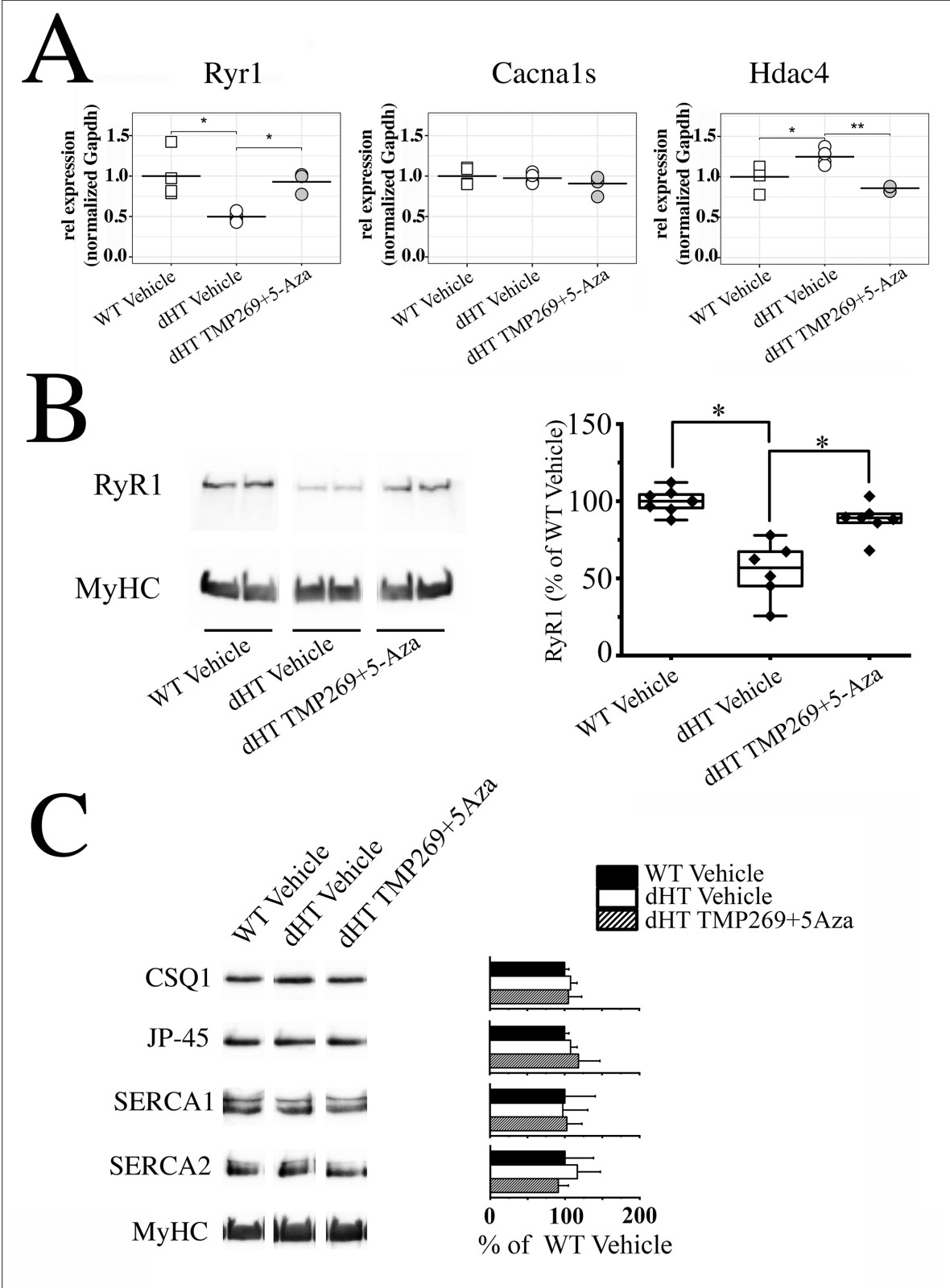

**Figure 5.** Treatment with TMP269 + 5-aza-2-deoxycytidine (5-Aza) reverses ryanodine receptor 1 (RyR1) loss in soleus muscles from double heterozygous (dHT) mice. (**A**) Real-time quantitative polymerase chain reaction (qPCR) on RNA isolated from soleus muscles isolated from vehicle-treated wild type (WT) and vehicle-treated and TMP269 + 5-Aza-treated dHT mice. Experiments were carried out on muscles isolated from 4 mice per group. RNA isolation and amplification conditions as described in the Materials and methods section. *p < 0.05; **p < 0.01 ANOVA followed by the Bonferroni post

*Figure 5 continued on next page*

*Figure 5 continued*

hoc test. (**B**) Western blot analysis of RyR1 content in total homogenates of soleus muscle from WT and dHT mice treated with vehicle or TMP269 + 5-Aza. Proteins were separated on a 6% SDS-PAGE, blotted overnight onto nitrocellulose. Left panel: representative images of blot probed with anti-RyR1 Ab followed by anti-MyHC Ab for normalization. Right panel: data points are expressed as Whisker plots *$p < 0.05$ (ANOVA followed by the Bonferroni post hoc test). (**C**) (Left panel) representative immunoblots of total homogenates soleus muscles from WT (vehicle) and dHT (vehicle and TMP269 + 5-Aza) mice probed with the indicated antibodies. (Right panel) bar histograms showing the mean ± SD (n = 4 mice per group) intensity of the immunopositive band, expressed as % of the intensity of the band in WT (vehicle-treated) mice. MyHC was used for loading protein normalization. Forty μg of protein were loaded per lane and proteins were separated on 7.5% or 10% SDS-PAG. Exact p values are given in the text.

The online version of this article includes the following source data for figure 5:

**Source data 1.** Treatment with TMP269 + 5-aza-2-deoxycytidine (5-Aza) reverses ryanodine receptor 1 (RyR1) loss in soleus muscles from double heterozygous (dHT) mice.

**Source data 2.** Treatment with TMP269 + 5-aza-2-deoxycytidine (5-Aza) reverses ryanodine receptor 1 (RyR1) loss in soleus muscles from double heterozygous (dHT) mice.

**Source data 3.** Treatment with TMP269 + 5-aza-2-deoxycytidine (5-Aza) reverses ryanodine receptor 1 (RyR1) loss in soleus muscles from double heterozygous (dHT) mice.

**Source data 4.** Treatment with TMP269 + 5-aza-2-deoxycytidine (5-Aza) does not affect CSQ1 content in soleus muscles from double heterozygous (dHT) mice.

**Source data 5.** Treatment with TMP269 + 5-aza-2-deoxycytidine (5-Aza) does not affect CSQ1 content in soleus muscles from double heterozygous (dHT) mice.

**Source data 6.** Treatment with TMP269 + 5-aza-2-deoxycytidine (5-Aza) does not affect JP-45 content in soleus muscles from double heterozygous (dHT) mice.

**Source data 7.** Treatment with TMP269 + 5-aza-2-deoxycytidine (5-Aza) does not affect JP-45 content in soleus muscles from double heterozygous (dHT) mice.

**Source data 8.** Treatment with TMP269 + 5-aza-2-deoxycytidine (5-Aza) does not affect sarco(endo)plasmic reticulum CaATPAse 1 (SERCA1) content in soleus muscles from double heterozygous (dHT) mice.

**Source data 9.** Treatment with TMP269 + 5-aza-2-deoxycytidine (5-Aza) does not affect sarco(endo)plasmic reticulum CaATPAse 1 (SERCA1) content in soleus muscles from double heterozygous (dHT) mice.

**Source data 10.** Treatment with TMP269 + 5-aza-2-deoxycytidine (5-Aza) does not affect sarco(endo)plasmic reticulum CaATPAse 2 (SERCA2) content in soleus muscles from double heterozygous (dHT) mice.

**Source data 11.** Treatment with TMP269+ 5-aza-2-deoxycytidine (5-AZa) does not affect sarco(endo)plasmic reticulum CaATPAse 2 (SERCA2) content in soleus muscles from double heterozygous (dHT) mice.

life of the affected individuals and his/her family members. To date there are no therapies available for congenital myopathies.

In order to target these unmet clinical needs we exploited the heterozygous mouse model we created carrying compound heterozygous RyR1 mutations (p.Q1970fsX16 + p.A4329D), for a preclinical study aimed at developing a therapeutic strategy for patients with congenital myopathies. In particular, we tested small molecules targeting epigenetic modifying enzymes such as class II HDACs and DNMT on male mice. The reason for performing experiments only on male mice is twofold: (i) their muscle mass is generally greater than that of females and (ii) we wanted to avoid variability linked to hormonal fluctuations. Though the cross-sectional area of fibers is larger in males than in females, the fiber type composition is similar (*Staron et al., 2000*), thus we are confident that the beneficial effects of the treatment with TMP269 + 5-Aza we observed on muscle function will occur in male and female mice alike. Here, we show that the combined treatment with TMP269 and 5-Aza-2 deoxycytidine have strong ergogenic effects on muscle functions, tested both in vivo and in vitro. The improvement of muscle performance is consistent with the increase of both peak calcium transients and RyR1 protein content in total muscle homogenates and is apparently not associated with changes in the fiber type composition or changes of the minimal Feret's diameter of muscle fibers. We are confident that this study provides proof of concept for a therapeutic strategy aimed to enhance muscle strength in patients affected by congenital myopathies linked to recessive *RYR1* mutations.

Our data shows that the improvement of muscle performance in vivo is mainly due to the amelioration of muscle strength in slow twitch fibers. The muscle ultrastructure demonstrates a partial rescue of the ECC machinery (more triads, less of them being incomplete) and of the associated mitochondria and may explain in part the rescue of muscle function. The elucidation of the exact molecular

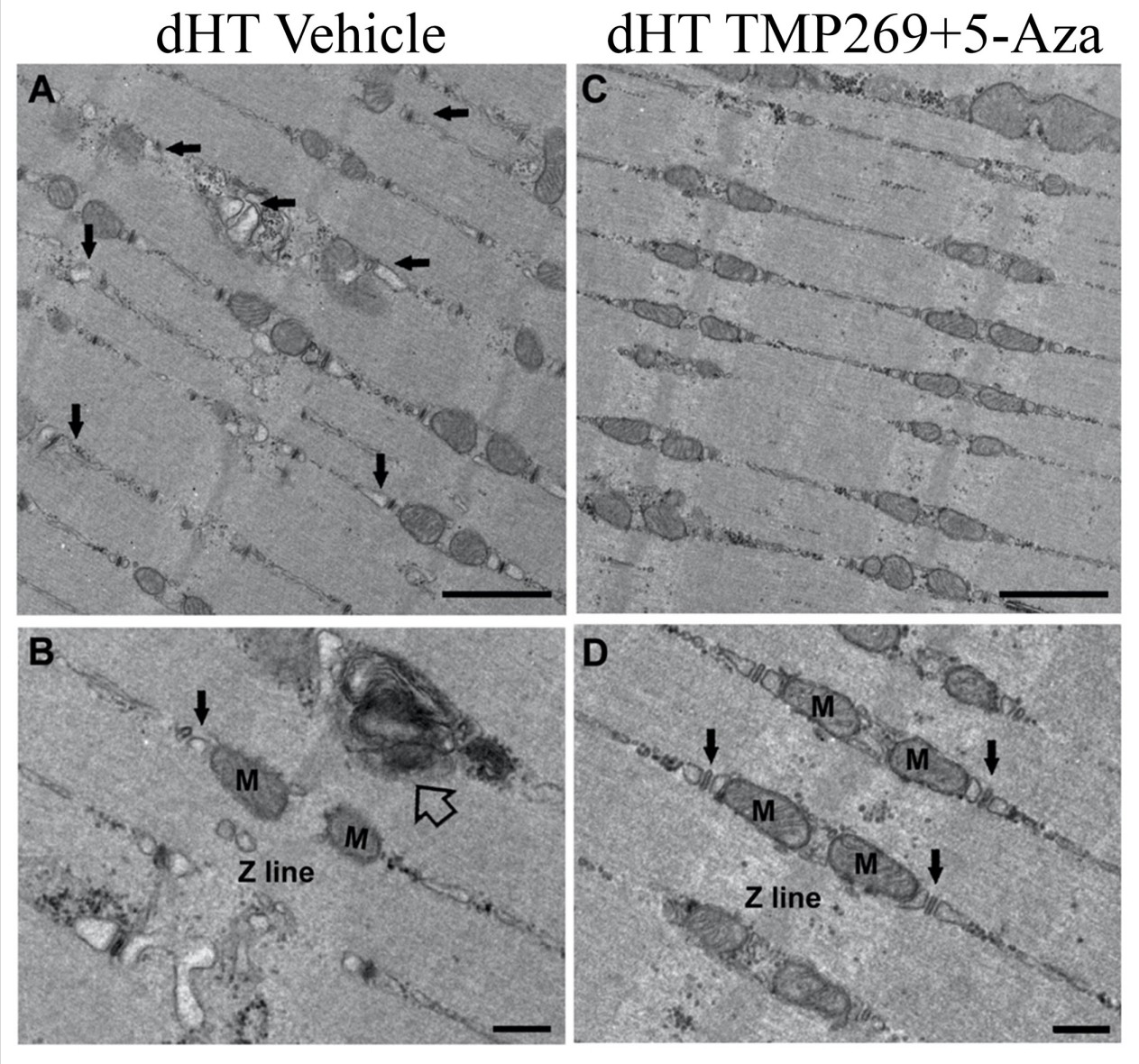

**Figure 6.** Representative electron microscopy (EM) images of soleus fibers from vehicle-treated (**A and B**) and TMP269 + 5-aza-2-deoxycytidine (5-Aza)-treated (**C and D**) double heterozygous (dHT) mice. (**A**) Representative EM images at low magnification of soleus fibers from vehicle-treated and (**C**) TMP269 + 5-Aza-treaded dHT mice: small arrows point to dilated sarcoplasmic reticulum (SR). (**B and D**) Higher magnification images showing calcium release units (CRUs, black arrow) and autophagic material (empty arrow, panel B) from vehicle-treated and drug-treated (**D**) dHT mice. M = mitochondria. Scale bars: A and C, 1 µm; B and D, 500 nm.

mechanisms underlying the improvement in strength specifically of slow twitch fibers caused by genome-wide effects of drugs inhibiting epigenetic modifying enzymes is an arduous challenge. However, we believe that a few assumptions can be taken into account to understand our data. First, histopathology of muscle biopsies from patients harboring *RYR1*-related congenital myopathies show fiber type I predominance (*Lawal et al., 2020*; *Jungbluth et al., 2005*; *Kondo et al., 2012*; *Bevilacqua et al., 2011*; *Lawal et al., 2018*). Thus, the recovery of muscle function after pharmacological treatment should occur to a greater extent in slow twitch fibers compared to fast twitch fibers, since the latter types of fibers are mostly absent. Increasing the strength of slow twitch fibers would greatly benefit patients. Furthermore, this interpretation of the histopathology of human skeletal muscle biopsies is consistent with our previous results on the dHT mouse model (*Elbaz et al., 2019*). Indeed, we found that the RyR1 array disarrangements and morphological alterations are more noticeable in fast glycolytic muscles such as EDLs compared to soleus muscles, a result in line with the idea that

**Table 1.** Quantitative analysis of calcium release units (CRUs).

In soleus fibers from TMP269 + 5-aza-2-deoxycytidine (5-Aza)-treated double heterozygous (dHT) mice, frequency of total CRUs and dyads (incomplete CRUs) are significantly rescued (columns A and B), suggesting a better preservation of CRUs and improved structure of triads.

| | A | B | C |
|---|---|---|---|
| | No. of CRUs /100 µm$^2$ | % of dyads | % of oblique longitudinal CRUs |
| Soleus Vehicle treated | 40.3 ± 2.4 | 14.2 ± 3.6 | 10.7 ± 2.0 |
| Soleus TMP269 + 5-Aza treated | [*]46.9 ± 2.3 | [†]6.5 ± 0.6 | 10.6 ± 2.6 |

Data are shown as mean ± SEM; n = 2 vehicle-treated soleus muscles; n=3 drug-treated soleus muscle.
[*]p = 0.0248.
[†]p = 0.0001.

skeletal muscle oxidative fibers are more resistant to the damaging effects linked to the expression of recessive *RYR1* mutations (*Elbaz et al., 2019*). If this is the case, then upon pharmacological treatment, slow oxidative fibers should recover their function to a greater extent than fast twitch glycolytic fibers. We found that HDAC4 transcript levels in soleus muscles from dHT mice treated with a combination of TMP269 + 5-Aza decreases to a value similar to that found in WT littermates. Though we don't have a clear explanation for this observation, it appears to be in line with the proposal that high levels of class II HDACs are indicative of muscle damage or age-related atrophy and reduced expression of class II HDACs is associated with an increased running endurance (*Potthoff et al., 2007*; *Consalvi et al., 2011*). Second, it was recently demonstrated that pharmacological inhibition of class II HDACs causes deacetylation not only of nuclear histones but also of other targets, including MyHC and PGC1alpha (*Luo et al., 2019*), an event which causes an increase of the cytoplasmic level of the PGC1alpha protein. Interestingly, PGC1alpha is a crucial regulator of oxidative metabolism and mitochondrial biogenesis in skeletal muscle fibers (*Handschin and Spiegelman, 2006*). We found that administration of class II HDAC and DNMT inhibitors to dHT mice induces an increase in the number of mitochondria in slow twitch fibers. This result is in agreement with the observed increase in PGC1alpha activity observed after treatment with class II specific HDAC inhibitors (*Luo et al., 2019*). Third, reconstitution of the RyR1 arrays in the junctional SR might restore the RyR1 retrograde signal, which is important for the activation of Ca$_v$1.1 Ca$^{2+}$ currents across T tubular membranes. Since the Ca$_v$1.1 to RyR1 ratio is lower in slow twitch muscles (*Delbono and Meissner, 1996*), reactivation of the Ca$_v$1.1 calcium current might occur to a larger extent in slow twitch muscles from dHT-treated mice compared to fast twitch muscles. Though Ca$_v$1.1 calcium currents are not relevant for skeletal muscle ECC, they have been implicated in a broad range of muscle functions, including fatigue resistance,

**Table 2.** Quantitative analysis of mitochondria.
Soleus fibers from TMP269 + 5-aza-2-deoxycytidine (5-Aza)-treated double heterozygous (dHT) mice show a significant improvement in frequency, disposition, and morphology of mitochondria.

| | A | B | C |
|---|---|---|---|
| | No. of mitochondria in 100 µm$^2$ | No. of severely altered mitochondria in 100 µm$^2$ (%) | No. of mitochondrion /CRU pairs in 100 µm$^2$ |
| Soleus Vehicle treated | 59.7 ± 2.9 | 3.2 ± 0.5 (6.2) | 26.9 ± 2.2 |
| Soleus Drug treated | [*]71.0 ± 3.5 | [†]2.9 ± 0.6 (3.7) | [‡]34.5 ± 2.3 |

Data are shown as mean ± SEM; n = 2 vehicle-treated soleus muscles; n=3 drug-treated soleus muscles.
[*]p = 0.0175.
[†]p = 0.0475.
[‡]p=0.0201.

protein synthesis, and calcium-dependent signaling pathways involved in the maintenance of proper skeletal muscle function (*Lee et al., 2015*).

In conclusion, the present study demonstrates that the combined pharmacological treatment with an FDA-approved DNMT inhibitor and TMP269 improves muscle strength and performance in a mouse model for recessive *RYR1* congenital myopathy. Further studies are needed to define the exact mechanism of the ergogenic effects of these genome-wide epigenetic modifying drugs; nevertheless, our results provide the proof of concept for the development for pharmacological treatment of patients with congenital myopathies linked to recessive *RYR1* mutations.

## Materials and methods
### Compliance with ethical standards
All experiments involving animals were carried out on 16- to 21-week-old male mice unless otherwise stated. Experimental procedures were approved by the Cantonal Veterinary Authority of Basel Stadt (BS Kantonales Veterinäramt Permit numbers 1728 and 2950). All experiments were performed in accordance with ARRIVE guidelines and Basel Stadt Cantonal regulations.

### Drug injection protocol
The class II HDAC inhibitor TMP269 was purchased from Selleckchem (S7324), the DNMT inhibitor 5-Aza was from Sigma-Aldrich (A3656), PEG300 was from Merck (8.17019), NMP was from Sigma-Aldrich (328634). Intraperitoneal injections (30-gauge needle) started at the sixth week of age and continued daily for 10–15 weeks. Mice received vehicle (PEG300 and NMP), TMP269 (25 mg/kg), 5-Aza (0.05 mg/kg), a combination of the two compounds (TMP26 + 5-Aza, 25 mg/kg and 0.05 mg/kg, respectively). TMP269 and 5-Aza were diluted in PEG300 (500 µl/kg) and NMP (250 µl/kg). The final volume of drug or vehicle injected per mouse was 750 µl/kg body weight.

### Pharmacokinetics analysis of TMP269
Six-week-old mice were injected intraperitoneally with 25 mg/kg body weight (n = 4) of TMP269, and blood and muscle tissue were collected at different time points post injection and analyzed. Blood (20 µl) was collected from the tail vein using lithium heparin coated capillaries (Minivette POCT 20 µl LH, Sarstedt, Germany). Samples were collected prior treatment (T0) and 1, 3, 6, 12, 24, 48 hr after TMP269 injection, transferred directly into autosampler tubes, and kept at –20°C until analyzed. For skeletal muscle samples, the following protocol was used. At T = 0 and at different time points after TMP269 injection (1, 3, 12, 24, 48 hr), mice were sacrificed and their skeletal muscles were isolated, flash frozen, and stored in liquid nitrogen. On the day of the analysis, muscle samples were homogenized using a tissue homogenizer (Mikro-Dismembrator S, Sartorius, Aubagne, France) for two sequences of 30 s each.

The levels of TMP269 in blood and in skeletal muscle were quantified by high pressure LC-MS/MS as described (*Duthaler et al., 2019*). Briefly, A Shimadzu LC system (Kyoto, Japan) was used, TMP269 and TMP195 (internal standard, IS) were analyzed by positive electrospray ionization and multiple reaction monitoring. Methanol containing IS (5 nM TMP195) was used to extract TMP269 from blood and muscle homogenate samples (20 µl blood/muscle plus 175 µl methanol). The samples were vortex mixed for approximately 30 s and centrifuged at 10°C, 3220 *g* for 30 min. Ten µl of supernatant were injected into the LC-MS/MS system. Calibration lines were prepared in blank mouse blood and covered a range of concentrations, from 0.25 to 500 nM TMP269. A linear regression between TMP269 to the IS peak area ratio and the nominal concentration was established with a weighting of $1/x2$ to quantify the TMP269 concentration in blood or muscle. The Analyst 1.6.2 software (AB Sciex, Concord, Canada) was used to operate the LC-MS/MS system and to analyze the data.

### Genotyping dHT mice and real-time qPCR
Mouse genotyping was performed on mouse genomic DNA, by polymerase chain reaction (PCR) amplification of *Ryr1* exon 36 and exon 91 using specifically designed primers (*Supplementary file 5*), as previously described (*Elbaz et al., 2019*). Quantitative real-time polymerase chain reaction (qPCR) was carried out on RNA extracted from muscles isolated from 21-week-old mice (treated with drug or vehicle for 15 weeks, starting at 6 weeks of age). Briefly, total RNA extracted from frozen hind limb

muscles was isolated, genomic DNA was removed by DNase I treatment (Invitrogen; 18068–015) and 1000 ng RNA were reverse-transcribed into cDNA using the High capacity cDNA Reverse Transcription Kit (Applied Biosystems; 4368814) as previously described (*Elbaz et al., 2019*). The cDNA was amplified by qPCR using the primers listed in *Supplementary file 5* and transcript levels were quantified using Power SYBR Green reagent Master Mix (Applied Biosystems; 4367659), using the Applied Biosystems 7500 Fast Real-time PCR System running 7500 software version 2.3 as previously described (*Elbaz et al., 2019*). Transcript quantification was based on the comparative $\Delta\Delta$Ct method. Each reaction was performed in duplicate and averaged and results are expressed as relative gene expression normalized to the housekeeping gene glyceraldehyde 3-phosphate dehydrogenase.

## In vivo muscle strength assessment

In vivo muscle performance was evaluated in male mice, by performing the following measurements: (i) forelimb grip strength and (ii) spontaneous locomotor activity. Forelimb grip strength was assessed once per week for a period of 10 weeks using a Grip Strength Meter from Columbus Instruments (Columbus, OH), following the manufacturer's recommendations. The grip force value obtained per mouse was calculated by averaging the average value of 5 measurements obtained on the same day on the same mouse. To avoid experimental bias during measurements of grip force, experimenters were blinded to the genotype and treatment of mice. For spontaneous locomotor activity, after 15 weeks of treatment with vehicle or TMP269 + 5-Aza, mice were individually housed in cages equipped with a running wheel carrying a magnet as previously described (*Elbaz et al., 2019*; *Mosca et al., 2013*). Wheel revolutions were registered by reed sensors connected to an I-7053D Digital-Input module (Spectra), and the revolution counters were read by a standard laptop computer via an I-7520 RS-485-to-RS-232 interface converter (Spectra). Digitized signals were processed by the "mouse running" software developed at Santhera Pharmaceuticals (*Briguet et al., 2004*). Total running distance (kilometer) and speed (km/h) were evaluated.

## Ex vivo muscle strength assessment

To test muscle force ex vivo, EDL and soleus muscles were dissected from 21-week-old male WT and dHT mice, after 15 weeks of treatment with vehicle alone, or with the combination of TMP26915-Aza. Isolated EDL and soleus muscles were mounted onto a muscle force transducing setup (MyoTronic, Heidelberg, Germany) as previously described (*Mosca et al., 2013*). Muscle force was digitized at 4 kHz by using an AD Instrument converter and stimulated with 15 V pulses for 1.0 ms. Tetanus was recorded in response to a train of pulses of 400 and 1100 ms duration delivered at 10/20/50/100/150 Hz, and 10/20/50/100/120 Hz, for EDL and soleus, respectively. Specific force was normalized to the muscle cross-sectional area [CSA_wet weight (mg)/length (mm)_1.06 (density mg/mm$^3$)] (*Brooks and Faulkner, 1988*). The experimenter performing the measurements was blinded with respect to the mouse genotype and treatment.

## Isolation of single FDB fibers for intracellular Ca$^{2+}$ measurements

Twenty-one-week-old WT and dHT vehicle- and drug-treated male mice were killed by pentobarbital overdose according to the procedures approved by the Kantonal Veterinary Authority. FDB muscles were isolated and digested with 0.2% of collagenase type I (*Clostridium hystoliticum* Type I, Sigma-Aldrich) and 0.2% of collagenase type II (*Clostridium hystoliticum* Type II, Worthington) in Tyrode's buffer (137 mM NaCl, 5.4 mM KCl, 0.5 mM MgCl$_2$, 1.8 mM CaCl$_2$, 0.1% glucose, 11.8 mM HEPES, pH 7.4 NaOH) for 45 min at 37°C as described (*Calderón et al., 2009*). Muscles were washed with Tyrode's buffer to block the collagenase activity and gently separated from tendons using large to narrowest set of fire-polished Pasteur pipettes. Fibers obtained by this procedure remained excitable and contracted briskly when assayed experimentally. Finally, fibers were placed on laminin coated (5 µl of 1 mg/ml mouse laminin from Thermo Fisher) 35 mm glass bottom dishes (MatTek corporation) for measurements of the resting [Ca$^{2+}$] or on ibiTreat 15µ-Slide four well (Ibidi) for electrically evoked Ca$^{2+}$ measurements as previously described (*Elbaz et al., 2019*).

## Measurements of resting [$Ca^{2+}$] and of electrically evoked $Ca^{2+}$ transients

Single FDB fibers were isolated from 4 to 5 male mice per group and allowed to adhere to laminin-treated 35 mm glass bottom dishes for 1 hr at 37°C. The fibers were then loaded with 5 µM Fura-2 AM (Invitrogen) by incubating them for 20 min at 19°C in Tyrode's buffer. The excess Fura-2 was diluted out by the addition of fresh Ringer's solution and measurements of the resting [$Ca^{2+}$] were carried out using an inverted Zeiss Axiovert fluorescent microscope, as previously described (*Elbaz et al., 2019*; *Mosca et al., 2013*). Only those fibers that contracted when an electrical stimulus was applied were used for the [$Ca^{2+}$] measurements.

For electrically evoked $Ca^{2+}$ transients, single FDB fibers were incubated for 10 min at 19°C in Tyrode's solution containing 10 µM low affinity calcium indicator Mag-Fluo-4 AM (Thermo Fisher), 50 µM *N*-benzyl-*p*-toluene sulfonamide (BTS, Tocris). Fibers were rinsed twice with fresh Tyrode's solution, and measurements were carried out in Tyrode's solution containing 50 µM BTS. Measurements were carried out with a Nikon Eclipse inverted fluorescent microscope equipped with a 20× PH1 DL magnification objective. The light signals originating from a spot of 1 mm diameter of the magnified image of FDB fibers were converted into electrical signals by a photomultiplier (Myotronic, Heidelberg, Germany). Fibers were excited at 480 nm and then stimulated either with a single pulse of 50 V with a duration of 1 ms or with a train of pulses of 50 V with a duration of 300 ms delivered at 100 Hz. Fluorescent signals were acquired using PowerLab Chart7. Changes in fluorescence were calculated as ΔF/F0 = (Fmax−Frest)/(Frest). Kinetic parameters were analyzed using Chart7 software. FDB fibers were isolated from 4 to 6 mice per group and results were averaged.

## Biochemical analysis of total muscle homogenates

Total muscle homogenates were prepared from soleus muscles isolated from WT and dHT vehicle- and drug-treated mice. SDS-polyacrylamide electrophoresis and Western blots of total homogenates were carried out as previously described (*Elbaz et al., 2019*; *Mosca et al., 2013*; *Schägger and von Jagow, 1987*). Western blots were stained with the primary antibodies listed in *Supplementary file 6*, followed by peroxidase-conjugated Protein G (Sigma P8170; 1:130,000) or peroxidase-conjugated anti-mouse IgG (Fab Specific) Ab (Sigma A2304; 1:200,000). The immunopositive bands were visualized by chemiluminescence using the WesternBright ECL- HRP Substrate (Witec AG). Densitometry of the immunopositive bands was carried out using the Fusion Solo S (Witec AG).

## Genome-wide DNA methylation analysis

Sodium bisulfite-treated DNA of nine samples (six dHT vehicle treated and three dHT drug treated) was subjected to measure global DNAm by *Infinium Mouse methylation* BeadChip array (Illumina) according to manufacturer's instructions. Illumina mouse array contains ~285,000 methylation sites per sample at single-nucleotide resolution covering over 20 design categories including gene promoters and enhancers. Downstream analysis were conducted using annotation developed previously mapping to mm10. Raw IDAT files were preprocessed using R/Bioconductor package SeSAMe (*Zhou et al., 2018*). Samples were normalized using 'noob' including background subtraction and dye-bias normalization. Methylation levels were computed as Beta (β) values (Meth/[Meth + Unmeth +100]) and as logit-transformed M-value (log2 [Meth/Unmeth]). Probes where at least one sample had a detection p value > 0.05 and the probes recommended previously using mm10 genome as general-purpose masking probes were filtered out (*Zhou et al., 2018*). The remaining 215,717 CpGs were used for further analysis. Statistical analyses were performed on M-values (*Du et al., 2010*) using bioconductor *limma*, whereas β-values were used for biological interpretation. Normalized methylation levels were evaluated using linear regression model between dHT treatment vs. vehicle. Differentially methylated CpGs were identified using empirical Bayes moderated t-statistics and associated Benjamini-Hochberg false discovery rate adjusted p values <0.05 were used as cutoff.

## Histological examination

Soleus muscles from treated and untreated WT and dHT mice were isolated and mounted for fluorescence microscopy imaging. Muscles were embedded in OCT and deep-frozen in 2-methylbutane, then stored at −80°C. Subsequently, transversal 10 µm sections were obtained using a Leica Cryostat (CM1950) starting from the belly of the muscles. Sections were stained as described by *Delezie et al.,*

*2019*, using the following primary antibodies (see *Supplementary file 6* for suppliers and catalog numbers): mouse IgG2b anti-MyHC I (1:50), mouse IgG1 anti-MyHC IIa (1:200), mouse IgM anti-MyHC IIb (1:100), and rabbit anti-mouse laminin (1:1500), followed by incubation with the following secondary antibodies: goat anti-mouse IgG1Alexa Fluor 568 (1:1000), goat anti-mouse IgM Alexa Fluor 488 (1:1000), goat anti-mouse IgG Fcγ2b Dylight 405 (1:400), and donkey anti-rabbit IgG Alexa Fluor 647 (1:2000). Images were obtained using an Eclipse Ti2 Nikon Fluorescence microscope with 10× air objective lens. Muscles from 3 mice per group were evaluated. Images were analyzed using Fiji plugins developed by *Delezie et al., 2019*, in order to obtain information on fiber types and minimal Feret's diameter (*Briguet et al., 2004*), the closest possible distance between the two parallel tangents of an object, using a combination of cell segmentation and intensity thresholds as described (*Briguet et al., 2004*).

## Preparation and quantitative analysis of samples by EM

Soleus muscles were dissected from sacrificed animals, pinned on a Sylgard dish, fixed at room temperature with 3.5% glutaraldehyde in 0.1 M NaCaCO buffer (pH 7.4), and stored in the fixative at 4°C (*Pietrangelo et al., 2015*). Fixed muscle was then post-fixed in a mixture of 2% $OsO_4$ and 0.8% $K_3Fe(CN)_6$ for 1–2 hr, rinsed with 0.1 M sodium cacodylate buffer with 75 mM $CaCl_2$, en-bloc stained with saturated uranyl acetate replacement, and embedded for EM in epoxy resin (Epon 812). Ultra-thin sections (~40 nm) were cut in a Leica Ultracut R microtome (Leica Microsystem, Austria) using a Diatome diamond knife (Diatome Ltd. CH-2501 Biel, Switzerland) and examined at 60 kV after double-staining with uranyl acetate replacement and lead citrate, with an FP 505 Morgagni Series 268D electron microscope (FEI Company, Brno, Czech Republic), equipped with Megaview III digital camera (Munster, Germany) and Soft Imaging System (Germany).

## Quantitative analyses by EM

Data contained in *Tables 1 and 2* were collected on soleus muscles from 21-week-old dHT mice, either vehicle or TMP269 + 5-Aza treated. In each sample, 10–20 fibers were analyzed. In each fiber 2–3 micrographs (all at the same magnification, 14 K, and of non-overlapping regions) were randomly collected from longitudinal sections.

Number of CRUs (*Table 1*, column A), number of mitochondria, number of severely altered mitochondria, and number of mitochondrion-CRU pairs (*Table 2*, columns A–C, respectively) were reported as average number/100 μm² (*Boncompagni et al., 2009*). In each EM image, we also determined the number of dyads, that is, incomplete triads (*Table 1*, column B), and of oblique/longitudinal CRUs (*Table 1*, column C), and expressed as percentages over the total number of CRUs. Mean and SEM were determined using GraphPad Prism (GraphPad Software, San Diego, CA). Statistically significant differences between groups were determined by the Student's t test (GraphPad Software, San Diego, CA) or by a chi-squared test (GraphPad Software, San Diego, CA). Values of $p < 0.05$ were considered significant.

## Statistical analysis

Statistical analysis was performed using the Student's unpaired t test for normally distributed values when two groups were being compared, or the ANOVA test followed by the Bonferroni post hoc test for multiple group comparisons. The Mann-Whitney U test was used when values were not normally distributed. $p < 0.05$ was considered significant.

## Additional information

### Funding

| Funder | Grant reference number | Author |
| --- | --- | --- |
| Swiss National Science Foundation | SNF 310030_184765 | Susan Treves |
| Swiss Muscle Foundation | FRSMM | Francesco Zorzato |

| Funder | Grant reference number | Author |
| --- | --- | --- |
| NeRAB | | Susan Treves |
| RYR1 Foundation | | Francesco Zorzato |

The funders had no role in study design, data collection and interpretation, or the decision to submit the work for publication.

## Author contributions
Alexis Ruiz, Conceptualization, Data curation, Formal analysis, Investigation, Methodology, Project administration, Validation, Writing - review and editing; Sofia Benucci, Faiza Noreen, Formal analysis, Investigation, Methodology, Writing - review and editing; Urs Duthaler, Formal analysis, Investigation, Methodology, Validation, Writing - review and editing; Christoph Bachmann, Data curation, Investigation, Methodology, Writing - review and editing; Martina Franchini, Formal analysis, Investigation, Writing - review and editing; Laura Pietrangelo, Investigation, Methodology, Writing - review and editing; Feliciano Protasi, Data curation, Formal analysis, Supervision, Writing - original draft, Writing - review and editing; Susan Treves, Conceptualization, Data curation, Formal analysis, Funding acquisition, Investigation, Project administration, Resources, Supervision, Validation, Writing - original draft, Writing - review and editing; Francesco Zorzato, Conceptualization, Data curation, Formal analysis, Funding acquisition, Investigation, Methodology, Project administration, Resources, Supervision, Validation, Visualization, Writing - original draft, Writing - review and editing

## Author ORCIDs
Urs Duthaler (iD) http://orcid.org/0000-0002-7811-3932
Susan Treves (iD) http://orcid.org/0000-0002-0007-9631
Francesco Zorzato (iD) http://orcid.org/0000-0002-8469-7065

## Ethics
This study was performed in strict accordance with the recommendations of the Basel Stadt Kantonal authorities. All animals were handled according to approved institutional animal care and use committee. The protocols were approved by the Kantonal Veterinary Authorities included in Licence permits numbers 1728 and 2950.

## Decision letter and Author response
Decision letter https://doi.org/10.7554/eLife.73718.sa1
Author response https://doi.org/10.7554/eLife.73718.sa2

## Additional files

### Supplementary files
• Appendix 2—figure 2—source data 1. Daily intraperitoneal (i.p.) injections with TMP269 + 5-aza-2-deoxycytidine (5-Aza) increase acetylation of Lys residues and of H3K9 in muscles from double heterozygous (dHT) mice.

• Appendix 2—figure 2—source data 2. Daily intraperitoneal (i.p.) injections with TMP269 + 5-aza-2-deoxycytidine (5-Aza) increase acetylation of Lys residues and of H3K9 in muscles from double heterozygous (dHT) mice. Wild type (WT) and dHT mice received a daily injection vehicle or 25 mg/kg TMP269+0.05 mg/kg 5-Aza. After 15 weeks of treatment, approximately 200 *flexor digitorum brevis* (FDBs) fibers per mouse were isolated and resuspended in cracking buffer. Protein extracts were loaded onto a Tris-Tricine gel, blotted onto nitrocellulose, and probed as indicated, with anti-acetyl-Lys antibodies, anti-H3K9acetyl antibodies, and anti-H3 antibodies (loading control). The three panels show the immunoreactive bands (arrows) that were used for the Boxplot analysis of *Appendix 2—figure 2*. The immunoreactivity of the H3K9acetyl positive band obtained in muscles from vehicle-treated WT mice was set to 100%.

• Appendix 2—figure 2—source data 3. Daily intraperitoneal (i.p.) injections with TMP269 + 5-aza-2-deoxycytidine (5-Aza) increase acetylation of Lys residues and of H3K9 in muscles from double heterozygous (dHT) mice. Wild type (WT) and dHT mice received a daily injection vehicle or 25 mg/kg TMP269+0.05 mg/kg 5-Aza. After 15 weeks of treatment, approximately 200 *flexor digitorum brevis* (FDBs) fibers per mouse were isolated and resuspended in cracking buffer. Protein extracts

were loaded onto a Tris-Tricine gel, blotted onto nitrocellulose, and probed as indicated, with anti-acetyl-Lys antibodies, anti-H3K9acetyl antibodies, and anti-H3 antibodies (loading control). The three panels show the immunoreactive bands (arrows) that were used for the Boxplot analysis of *Appendix 2—figure 2*. The immunoreactivity of the H3K9acetyl positive band obtained in muscles from vehicle-treated WT mice was set to 100%.

• Appendix 2—figure 2—source data 4. Daily intraperitoneal (i.p.) injections with TMP269 + 5-aza-2-deoxycytidine (5-Aza) increase acetylation of Lys residues and of H3K9 in muscles from double heterozygous (dHT) mice. Wild type (WT) and dHT mice received a daily injection vehicle or 25 mg/kg TMP269+0.05 mg/kg 5-Aza. After 15 weeks of treatment, approximately 200 *flexor digitorum brevis* (FDBs) fibers per mouse were isolated and resuspended in cracking buffer. Protein extracts were loaded onto a Tris-Tricine gel, blotted onto nitrocellulose, and probed as indicated, with anti-acetyl-Lys antibodies, anti-H3K9acetyl antibodies, and anti-H3 antibodies (loading control). The three panels show the immunoreactive bands (arrows) that were used for the Boxplot analysis of *Appendix 2—figure 2*. The immunoreactivity of the H3K9acetyl positive band obtained in muscles from vehicle-treated WT mice was set to 100%.

• Appendix 2—figure 2—source data 5. Daily intraperitoneal (i.p.) injections with TMP269 + 5-aza-2-deoxycytidine (5-Aza) increase acetylation of Lys residues and of H3K9 in muscles from double heterozygous (dHT) mice. Wild type (WT) and dHT mice received a daily injection vehicle or 25 mg/kg TMP269+0.05 mg/kg 5-Aza. After 15 weeks of treatment, approximately 200 *flexor digitorum brevis* (FDBs) fibers per mouse were isolated and resuspended in cracking buffer. Protein extracts were loaded onto a Tris-Tricine gel, blotted onto nitrocellulose, and probed as indicated, with anti-acetyl-Lys antibodies, anti-H3K9acetyl antibodies, and anti-H3 antibodies (loading control). The three panels show the immunoreactive bands (arrows) that were used for the Boxplot analysis of *Appendix 2—figure 2*. The immunoreactivity of the H3K9acetyl positive band obtained in muscles from vehicle-treated WT mice was set to 100%.

• Supplementary file 1. List of hypomethylated protein-encoding genes in soleus muscles from double heterozygous (dHT) mice treated for 16 weeks with TMP269 + 5-aza-2-deoxycytidine (5-Aza) drug vs. vehicle-treated dHT mice.

• Supplementary file 2. Specific force of *extensor digitorum longus* (EDL) and soleus muscle from wild type (WT) and double heterozygous (dHT) mice treated with vehicle or TMP269 + 5-aza-2-deoxycytidine (5-Aza) for 16 weeks.

• Supplementary file 3. Fiber type composition of soleus muscles from mice treated for 16 weeks with vehicle or TMP269 + 5-aza-2-deoxycytidine (5-Aza).

• Supplementary file 4. Analysis of electrically evoked calcium transients in single *flexor digitorum brevis* (FDB) muscle fibers isolated from wild type (WT) and double heterozygous (dHT) littermates, treated with vehicle or TMP269 + 5-aza-2-deoxycytidine (5-Aza) (25 mg/kg) for 16 weeks.

• Supplementary file 5. Sequence of primers used and targets.

• Supplementary file 6. List of antibodies and suppliers, see *Zorzato et al., 2000* for JP-45.

• Transparent reporting form

## Data availability

All data, code, and materials used in the analysis are available in some form to any researcher for purposes of reproducing or extending the analysis. There are no restrictions on materials, such as materials transfer agreements (MTAs). All data are available in the main text or the supplementary materials.

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

## Appendix 1

The pharmacokinetics and bio-distribution of TMP269 confirm that i.p. injection of 25 mg/kg body weight of TMP269 dissolved in PEG300 (500 µl/kg) and NMP (250 µl/kg) results in accumulation of the drug in blood and skeletal muscles. Samples were collected at different time points and the content of TMP269 was quantified by LC-MS/MS (*Duthaler et al., 2019*). The peak blood concentration of TMP269 was achieved approximately 1 hr after injection. *Appendix 1—figure 1A* shows that the circulating levels of TMP269 decay within 12 hr. Importantly, the class II HDAC inhibitor diffuses into skeletal muscle (*Appendix 1—figure 1B*), and as expected, its concentration profile in skeletal muscle follows that observed in blood.

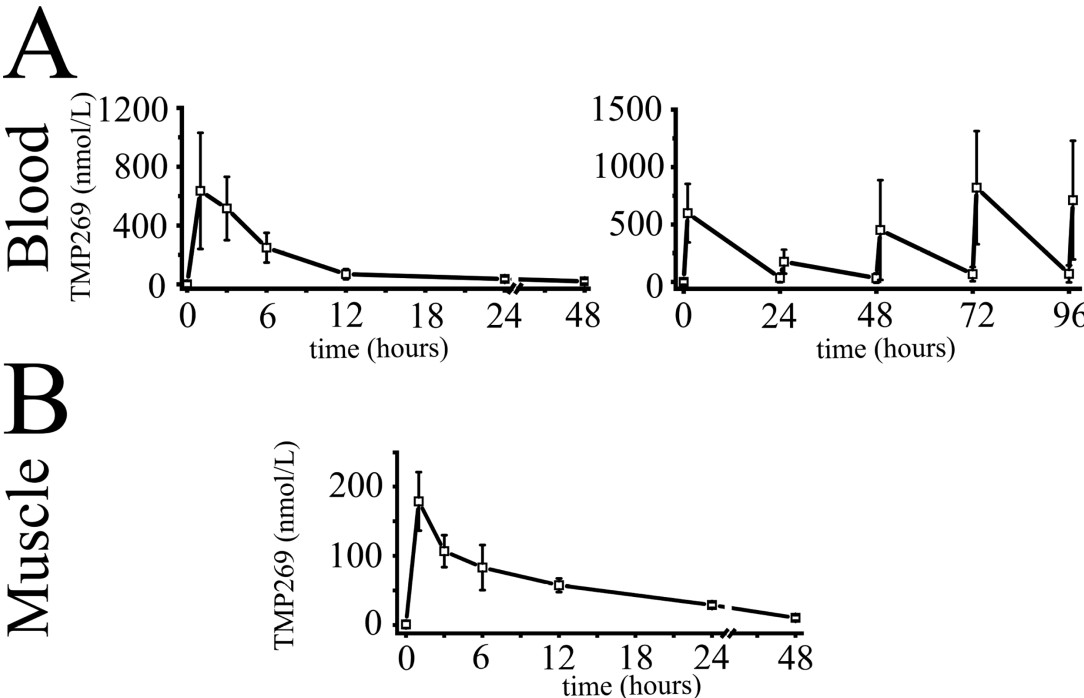

**Appendix 1—figure 1.** Pharmacokinetic profile of TMP269 following intraperitoneal injection in wild type (WT) mice.
 (**A**) Concentration-time course of TMP269 in blood. Left panel: mean (± SD, n = 4 mice) plasma concentration in mice receiving a single dose of 25 mg/kg of TMP269. Right panel: mean (± SD, n = 4 mice) blood concentration of TMP269 in mice after receiving consecutive doses of 25 mg/kg of TMP269 at t = 0, 24, 48, 72, and 96 hr. Blood samples were taken 10 min before and 1 hr after each intraperitoneal injection. (**B**) Mean (± SD; n = 4) TMP269 concentration in skeletal muscle after a single intraperitoneal injection of 25 mg/kg TMP269, during 48 hr.

## Appendix 2

This section confirms the biological effect of daily i.p. injections of TMP269 + 5-Aza since 165 protein-coding genes were hypomethylation in soleus muscles. Gene Ontology analysis of the hypomethylated genes shows that most of the hypomethylated genes belong to pathways involved in gene transcription, kinase activity, and membrane targeting (*Appendix 2—figure 1*). Furthermore, administration of TMP269 + 5-Aza increases the acetylation of Lys residues (*Appendix 2—figure 2A* and *Appendix 2—figure 2—source data 1–5*) and of H3K9 (*Appendix 2—figure 2B and D* and *Appendix 2—figure 2—source data 1–5*) in total homogenates from FDB fibers isolated from WT and dHT mice, compared to that observed in fibers from vehicle-treated WT and dHT mice. This result unequivocally indicates that the inhibition of the deacetylation activity of class IIa HDACs by TMP269 occurs in the nuclei.

# Gene Ontology Analysis

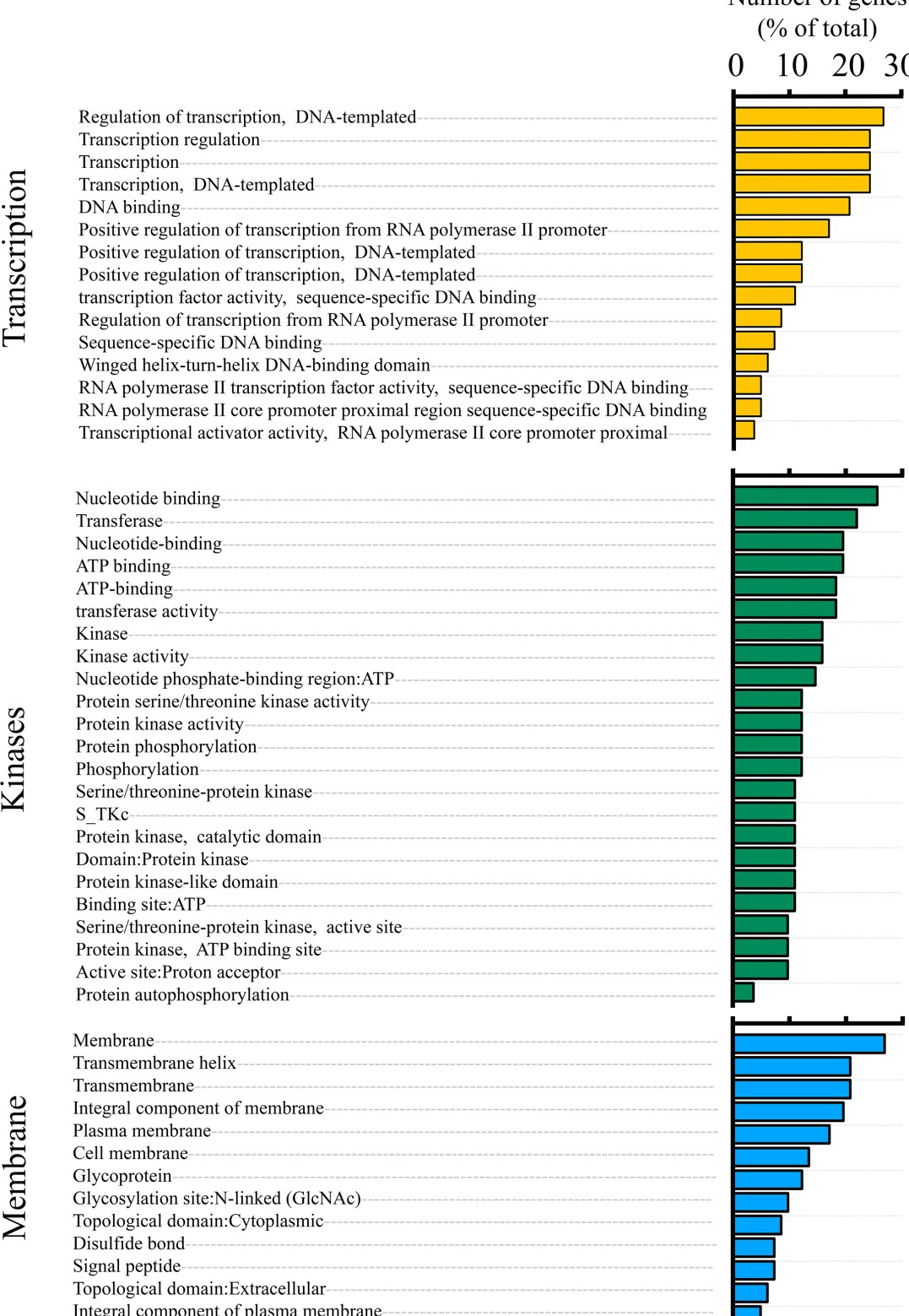

**Appendix 2—figure 1.** Gene Ontology (GO) analysis of hypomethylated genes in soleus muscles from double heterozygous (dHT) mice treated for 15 weeks with TMP269 + 5-Aza drug vs. vehicle-treated dHT mice. Genes showing a ≥2.0-fold decrease in methylation (p ≤ 0.05) were analyzed by DAVID functional annotation to produce

*Appendix 2—figure 1 continued on next page*

gene clusters (≥2 genes/cluster) corresponding to 51 GO annotation terms. GO terms corresponding to biological process (GOTERM_BP_FAT and KEGG_PATHWAY) were extracted and are plotted with the numbers of genes (as a percentage of the total) for each term. GO terms with <2% of the total genes were not plotted unless significantly enriched (Benjamini ≤0.05).

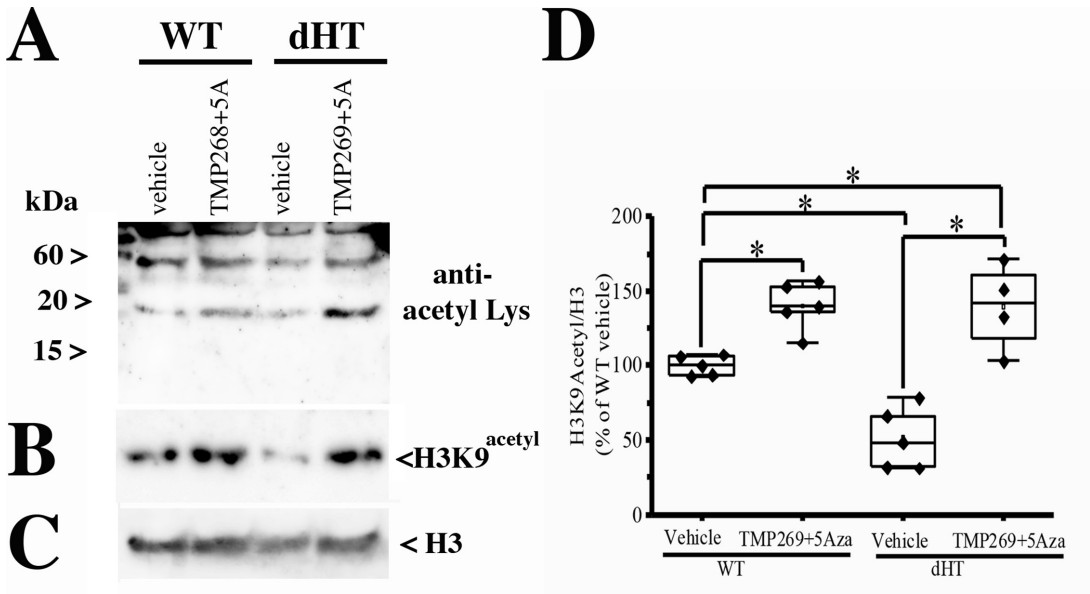

**Appendix 2—figure 2.** Daily intraperitoneal (i.p.) injections with TMP269 + 5-aza-2-deoxycytidine (5-Aza) increase acetylation of Lys residues and of H3K9 in muscles from double heterozygous (dHT) mice. Wild type (WT) and dHT mice received a daily injection vehicle or 25 mg/kg TMP269 + 0.05 mg/kg 5-Aza. After 15 weeks of treatment, approximately 200 *flexor digitorum brevis* (FDBs) fibers per mouse were isolated and resuspended in cracking buffer (10% glycerol, 5% β-mercaptoethanol, 2.5% SDS, 62.5 mM Tris pH 6.8, 6 M urea). Protein extracts were loaded on a Tris-Tricine gel, blotted onto nitrocellulose and probed with (**A**) anti-acetyl-Lys and (**B**) anti-H3K9$^{acetyl}$ antibodies. (**C**) Shows the immunoreactivity of anti-H3 (histone 3) for loading control. (**D**) Boxplot analysis of Western blots using anti-H3K9$^{acetyl}$ antibodies. The immunoreactivity of the H3K9$^{acetyl}$ positive band obtained in muscles from vehicle-treated WT mice was set to 100%. Each symbol represents the result obtained from a single mouse. *$p < 0.05$ (ANOVA followed by the Bonferroni post hoc test). WT vehicle vs. WT TMP269 + 5-Aza, p = 0.041; WT vehicle vs. dHT vehicle, p = 0.037; WT vehicle vs. dHT TMP269 + 5-Aza, p = 0.042; dHT vehicle vs. dHT TMP269 + 5-Aza, p = 0.021.

The online version of this article includes the following source data for appendix 2—figure 2:

- **Appendix 2—figure 2—source data 1.** Daily intraperitoneal (i.p.) injections with TMP269 + 5-aza-2-deoxycytidine (5-Aza) increase acetylation of Lys residues and of H3K9 in muscles from double heterozygous (dHT) mice.

- **Appendix 2—figure 2—source data 2.** Daily intraperitoneal (i.p.) injections with TMP269 + 5-aza-2-deoxycytidine (5-Aza) increase acetylation of Lys residues and of H3K9 in muscles from double heterozygous (dHT) mice.

- Wild type (WT) and dHT mice received a daily injection vehicle or 25 mg/kg TMP269+0.05 mg/kg 5-Aza. After 15 weeks of treatment, approximately 200 *flexor digitorum brevis* (FDBs) fibers per mouse were isolated and resuspended in cracking buffer. Protein extracts were loaded onto a Tris-Tricine gel, blotted onto nitrocellulose, and probed as indicated, with anti-acetyl-Lys antibodies, anti-H3K9acetyl antibodies, and anti-H3 antibodies (loading control). The three panels show the immunoreactive bands (arrows) that were used for the Boxplot analysis of *Appendix 2—figure 2*. The immunoreactivity of the H3K9acetyl positive band obtained in muscles from vehicle-treated WT mice was set to 100%.

- **Appendix 2—figure 2—source data 3.** Daily intraperitoneal (i.p.) injections with TMP269 + 5-aza-2-deoxycytidine (5-Aza) increase acetylation of Lys residues and of H3K9 in muscles from double

heterozygous (dHT) mice.

• Wild type (WT) and dHT mice received a daily injection vehicle or 25 mg/kg TMP269+0.05 mg/kg 5-Aza. After 15 weeks of treatment, approximately 200 *flexor digitorum brevis* (FDBs) fibers per mouse were isolated and resuspended in cracking buffer. Protein extracts were loaded onto a Tris-Tricine gel, blotted onto nitrocellulose, and probed as indicated, with anti-acetyl-Lys antibodies, anti-H3K9acetyl antibodies, and anti-H3 antibodies (loading control). The three panels show the immunoreactive bands (arrows) that were used for the Boxplot analysis of *Appendix 2—figure 2*. The immunoreactivity of the H3K9acetyl positive band obtained in muscles from vehicle-treated WT mice was set to 100%.

• **Appendix 2—figure 2—source data 4.** Daily intraperitoneal (i.p.) injections with TMP269 + 5-aza-2-deoxycytidine (5-Aza) increase acetylation of Lys residues and of H3K9 in muscles from double heterozygous (dHT) mice.

• Wild type (WT) and dHT mice received a daily injection vehicle or 25 mg/kg TMP269+0.05 mg/kg 5-Aza. After 15 weeks of treatment, approximately 200 *flexor digitorum brevis* (FDBs) fibers per mouse were isolated and resuspended in cracking buffer. Protein extracts were loaded onto a Tris-Tricine gel, blotted onto nitrocellulose, and probed as indicated, with anti-acetyl-Lys antibodies, anti-H3K9acetyl antibodies, and anti-H3 antibodies (loading control). The three panels show the immunoreactive bands (arrows) that were used for the Boxplot analysis of *Appendix 2—figure 2*. The immunoreactivity of the H3K9acetyl positive band obtained in muscles from vehicle-treated WT mice was set to 100%.

• **Appendix 2—figure 2—source data 5.** Daily intraperitoneal (i.p.) injections with TMP269 + 5-aza-2-deoxycytidine (5-Aza) increase acetylation of Lys residues and of H3K9 in muscles from double heterozygous (dHT) mice.

• Wild type (WT) and dHT mice received a daily injection vehicle or 25 mg/kg TMP269+0.05 mg/kg 5-Aza. After 15 weeks of treatment, approximately 200 *flexor digitorum brevis* (FDBs) fibers per mouse were isolated and resuspended in cracking buffer. Protein extracts were loaded onto a Tris-Tricine gel, blotted onto nitrocellulose, and probed as indicated, with anti-acetyl-Lys antibodies, anti-H3K9acetyl antibodies, and anti-H3 antibodies (loading control). The three panels show the immunoreactive bands (arrows) that were used for the Boxplot analysis of *Appendix 2—figure 2*. The immunoreactivity of the H3K9acetyl positive band obtained in muscles from vehicle-treated WT mice was set to 100%.

## Appendix 3

The baseline Mag-Fluo-4 fluorescence (in arbitrary units) is similar in FDBs from vehicle-treated WT mice (n = 91 fibers isolated from n = 4 mice), from vehicle-treated dHT mice (n = 110 fibers isolated from n = 6 mice) and from drug-treated dHT mice (n = 155 fibers isolated from n = 5 mice).

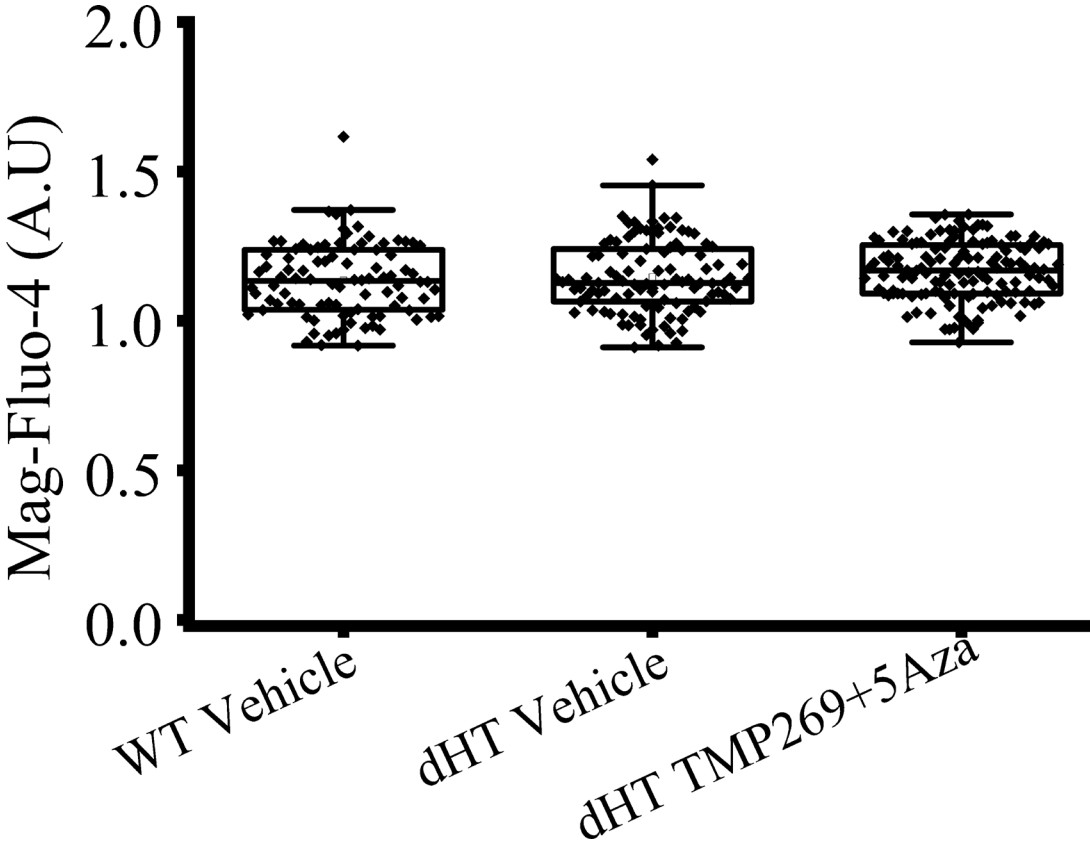

**Appendix 3—figure 1.** Resting fluorescence values in *flexor digitorum brevis* (FDB) fibers loaded with 10 μM Mag-Fluo-4. Raw resting fluorescence values (arbitrary units, AU) were obtained from the light signal originating from a spot of 1 mm diameter of the magnified image of the FDB fiber. The light was converted into an electrical signal by a photomultiplier connected to a Nikon Eclipse TL200-E Fluorescent microscope (Nikon Instruments Inc, Amsterdam, The Netherlands) with a 20× Plan Apo VC Nikon Objective (1.4 NA). Fluorescent signals were acquired using PowerLab Chart7. Each symbol represents the fluorescence value from one FDB fiber.

