## [Editor Report]

The paper describes improvement in muscle phenotype of a congenital myopathy mouse model by a combined treatment with pharmacological inhibitors of Class IIa histone deacetylases and DNA methylases. The paper demonstrates in principle that there are treatment avenues to pursue but their application could be limited as phenotypic rescue appears to be restricted to particular muscle fiber types.

---

## [Decision Letter]

**Decision letter after peer review:**

Thank you for submitting your article "Improvement of muscle strength in a mouse model for congenital myopathy treated with HDAC and DNA methyltransferase inhibitors" for consideration by *eLife*. Your article has been reviewed by 3 peer reviewers, one of whom is a member of our Board of Reviewing Editors, and the evaluation has been overseen by Mone Zaidi as the Senior Editor. The following individual involved in review of your submission has agreed to reveal their identity: Jianjie Ma (Reviewer #3).

We thus report a very positive response to your paper, and very much hope you will be able to prepare a revised version. We received very positive comments from all three reviewers.

Essential revisions:

(A) Reviewer 1 does not have major requests to make and states that "I do not have major comments, subject to the views of the remaining reviewers within their own areas of expertise bearing on particular aspects of the paper".

(B) Reviewer 2 has the following revision requests:

(1) Concerning the manuscript:

(a) It appears that the ca^2+^ imaging with Mag-Fluo4 were not calibrated for dye loading beyond normalising to basal fluorescence. Could the authors comment on steps taken to ensure similar dye loading?

(b) The discussion refers to correlations between HDAC4 expression levels and *MEF2*-dependent gene expression- is expression of any *MEF2* target genes altered by TMP269+Aza treatment?

(2) Concerning requirements for further experimental data, "….that the study would benefit from additional simple mechanistically relevant experiments."

(a) In Figure 1 average muscle strength data is plotted for each genotype group of mice and it is not clear from this data representation whether all mice show improvement in muscle strength. Data for individual mice before and after treatment would make a compelling case for improved muscle function in all individuals.

(b) It is not clear whether the ca^2+^ imaging with Mag-Fluo4 was calibrated for dye loading.

(c) The inhibitors appear beneficial only for slow oxidative soleus fibers, the implications of this for treatment of myopathy in humans could be discussed.

(d) The authors have examined HDAC4 expression levels but a key determinant of Class IIa HDAC function is their subcellular localization -in the nucleus they ac as *MEF2* repressors and it would be informative to know whether HDAC4 is more nuclear in the dHT mice and whether TMP269+Aza treatment alters its localisation.

(C) Reviewer 3 has the following revision requests:

(1) Concerning the manuscript:

(a) The longitudinal studies with drug treatment of the knock-in mice are well-conducted. It appears that all the experiments were conducted with male mice. The authors may want to discuss how sex may impact the conclusion.

(b) The authors conducted genome-wide DNA methylation analysis and identified a group of genes that underwent hypo-methylation with the treatment of TMP269 and 5-Aza-2-deoxycytidine. It would be helpful if gene ontology (GO) analyses can be used to draw inference on the signaling network associated with the changes in calcium signaling machinery in the background of RYR1 mutation.

(c) While the DNA methylation studies can provide useful information, complementary studies may be performed using RNAseq analysis to quantify the changes in the mRNA of the genes that could underlie the improved contractile function in the mutant mice following drug treatment.

---

## [Author Response]

Essential revisions:(A) Reviewer 1 does not have major requests to make and states that "I do not have major comments, subject to the views of the remaining reviewers within their own areas of expertise bearing on particular aspects of the paper".

We would like to thank Reviewer 1 for his/her positive feedback.

(B) Reviewer 2 has the following revision requests:(1) Concerning the manuscript:(a) It appears that the ca^2+^ imaging with Mag-Fluo4 were not calibrated for dye loading beyond normalising to basal fluorescence. Could the authors comment on steps taken to ensure similar dye loading?

In order to ensure similar Mag-Fluo4 loading into FDB fibers we carefully controlled time and temperature during the dye loading incubation. Briefly, FBD fibers were first placed in a Warner Instruments RC-27NE chamber mounted in a P6 platform and kept for 15 min in an incubator at 19.0° C. Then the buffer solution in the chamber was replaced with Tyrode's containing 10 M MagFluo4-AM and the FDB fibers were incubated with the new solution for 10 min at 19.0° C. The resulting mean resting MagFluo-4 fluorescence in FDB fibers from WT and dHT mice was similar. Since, the average resting [ca^2+^] as determined with fura-2 was similar in FDB from WT and dHT mice (page 13, lines 248-252) we assume that there are no major differences in the amount of dye loaded into FDB fibers from vehicle treated WT, dHT and drug treated dHT mice.

We have added a new figure (Appendix 3-figure 1) showing the resting Mag-Fluo-4 fluorescence level (arbitrary units) in FDB fibers. We have also added a sentence commenting on the similar dye loading in the revised manuscript (page 13 lines 252-258).

(b) The discussion refers to correlations between HDAC4 expression levels and MEF2-dependent gene expression- is expression of any MEF2 target genes altered by TMP269+Aza treatment?

In the discussion of the unrevised manuscript, we pointed out that it is an arduous challenge to define the exact mechanism underlying the effect of drugs inhibiting epigenetic modifying enzymes. On the basis of data present in the literature, we speculated on potential explanations which may account for the selective effect on slow twitch muscles of TMP269/Aza treatment. In the revised manuscript we deleted the sentence referring to the speculation concerning MEF-2 dependent gene expression (lines 396 and 397 of the unrevised manuscript). Prompted by the reviewer's comment, we performed new RT-PCR experiments on Soleus muscles from mice treated for 15 weeks with vehicle or drugs, and we analysed the expression of Ryr1 together withSlc2a4 (encoding Glut4), a gene which has been reported to be MEF-2 dependent (Liu et al.,1994 J. Biol. Chem 269: 28514-28521).

As shown in Author response image 1, soleus muscles from dHT exhibit a decrease of expression of GLUT4 (gene Slc2a4, right panel). At variance with the Ryr1 transcript (left panel), the expression of GLUT4 does not exhibit a statistical significant increase upon TMP269+5-Aza treatment.

**Author response image 1. sa2fig1:** qPCR of Ryr1 and Slc2a4 expression levels in soleus muscles isolated from WT and dHT mice treated for 15 weeks with vehicle or TMP269+5-Aza. Each symbol represents the value (mean of duplicate) from a single mouse.

(2) Concerning requirements for further experimental data, "….that the study would benefit from additional simple mechanistically relevant experiments."(a) In Figure 1 average muscle strength data is plotted for each genotype group of mice and it is not clear from this data representation whether all mice show improvement in muscle strength. Data for individual mice before and after treatment would make a compelling case for improved muscle function in all individuals.

In the new Figure 1A—figure supplement 1 we show the raw data pertaining to the grip strength of each vehicle and drug treated dHT mouse before and after 10 weeks of drug treatment. Although there was some variability in the response between the dHT mice which were tested, Figure 1A-supplement 1 clearly shows that in each dHT mouse the treatment with TMP269+5-Aza induces an increase of grip strength. This information has also been added in the revised text (page 7 lines 154-156 of the revised manuscript).

(b) It is not clear whether the ca^2+^ imaging with Mag-Fluo4 was calibrated for dye loading.

See answer to point a) of reviewer 2

(c) The inhibitors appear beneficial only for slow oxidative soleus fibers, the implications of this for treatment of myopathy in humans could be discussed.

Histopathological findings on muscle biopsies from patients with RYR1-related congenital myopathies show fiber type 1 predominance (Kondo et al., 2012 Am J Med Genet. 158A:772-778; Bevilacqua et al., 2011 Neuropath Appl Neurobiol 37: 271-284; Lawal et al., 2018 Neurotherapeutics 15:885-899). Thus, the recovery of muscle function after pharmacological treatment should occur to a greater extent in slow twitch fibers compared to fast twitch fibers, since the latter types of fibers are mostly absent. We have added a sentence in the discussion of the revised manuscript (page 22, lines 420-424) to emphasize the potential relevance of our preclinical study for the treatment of patients affected by congenital myopathies linked to recessive *RYR1* mutations.

(d) The authors have examined HDAC4 expression levels but a key determinant of Class IIa HDAC function is their subcellular localization -in the nucleus they ac as MEF2 repressors and it would be informative to know whether HDAC4 is more nuclear in the dHT mice and whether TMP269+Aza treatment alters its localisation.

Appendix 2-Figure 2 of the revised manuscript (Figure S2 of the unrevised manuscript) shows that TMP269+5-Aza treatment rescues the acetylation of H3K9, a histone protein associated with nuclear chromatin. This result unequivocally indicates that the inhibition of the deacetylation activity of class IIa HDACs by TMP269 occurs in the nuclei. In addition, Supplementary Figure S3 also shows that in the absence of TMP269+5-Aza treatment, the level of H3K9 acetylation in FDBs from vehicle -treated dHT mice is 50% lower compared to FDBs from WT mice. Histone H3 is one of the main histone proteins involved in the structure of chromatin in eukaryotic cells and its acetylation is linked to active transcription.

The results shown in Appendix 2-Figure 2 indicate that deacetylation activity linked to class IIa HDACs is higher in vehicle treated dHT mice. Assuming that the enzymatic turn-over rate of class IIa HDACs from WT is similar to that of dHT mice, the increase of deacetylation activity in dHT is consistent with an enrichment of HDAC proteins in the nuclei of dHT. The recovery of the H3K9 acetylation brought about by TMP269+5-Aza treatment reflects the inhibitory effect of TMP269 on the enzymatic activity of the class IIa HDACs rather than alteration of the HDAC4 protein trafficking, because to our knowledge there are no data showing that TMP269 stimulates calmodulin-dependent kinase 2 whose phosphorylation activity promotes nuclear efflux of HDAC4. We added a sentence in the revised manuscript to highlight the results of Figure S3 (page 6, lines 126- 133).

(C) Reviewer 3 has the following revision requests:(1) Concerning the manuscript:(a) The longitudinal studies with drug treatment of the knock-in mice are well-conducted. It appears that all the experiments were conducted with male mice. The authors may want to discuss how sex may impact the conclusion.

The reason for performing experiments only on male mice was twofold: (i) their muscle mass and force development is generally greater than that of females and (ii) we wanted to avoid variability linked to hormonal fluctuations. Although the cross sectional area of muscle fibers of female mice is smaller compared to that of male mice, there is no differences in fiber type composition (Staron et al., 2000 J Histochem Cytochem 48: 623-629). Thus, assuming the fraction of slow twitch fibers within skeletal muscle is the same between male and female we are confident that the therapeutic effect of TMP269+5-Aza treatment will occur also in females. We have added a sentence on the revised version of the manuscript (page 21, lines 399-404).

(b) The authors conducted genome-wide DNA methylation analysis and identified a group of genes that underwent hypo-methylation with the treatment of TMP269 and 5-Aza-2-deoxycytidine. It would be helpful if gene ontology (GO) analyses can be used to draw inference on the signaling network associated with the changes in calcium signaling machinery in the background of RYR1 mutation.

We have added a supplementary Figure (Appendix 2-Figure 1) to the revised manuscript showing GO analysis of hypomethylated genes in soleus muscles from dHT mice treated for 15 weeks with TMP269+5-Aza drug versus vehicle treated dHT mice (page 5, lines 111-113).

(c) While the DNA methylation studies can provide useful information, complementary studies may be performed using RNAseq analysis to quantify the changes in the mRNA of the genes that could underlie the improved contractile function in the mutant mice following drug treatment.

We agreed that RNAseq analysis would be interesting and may help elucidate the mechanism leading to the improved muscle function in drug treated dHT mice. Preliminary TMT mass spectrometry analysis of muscles from dHT and WT before treatment suggest that there are a large number of proteins (>3000) in EDL and Soleus muscles exhibiting differential expression in dHT compared to WT. The full characterization and validation of these data sets represents a new project in itself. We think it is goes beyond the goal of the presents study to perform RNAseq in muscles from drug and vehicle treated WT and dHT mice.